EMBO
Molecular Medicine

# Immune training enhances anti-viral responses and improves outcomes in *Pax5*−/+ mice susceptible to chronic infection

Zhe Lu[1,2,17], Olivia Stencel[1,2,17], Wei Liu[2,3,4], Eleni Vasileiou [1,2,3,4], Haifeng C Xu[5], Piyush Pandey[5], Paweł Stachura [1,2,5], Abdelrahman Elwy[6], Anastassia Tsombal[1,2], Ann-Sophie Mai[2,3,4], Franziska Auer[7], Mina N F Morcos [7], Maximilian Seidl[8], Sarah Koziel [2,9,10,11], Peter-Martin Bruch[2,9,10,11], Sascha Dietrich[2,9,10,11], Sarah Elitzur [12], Gunther Hartmann[3,4], Karl S Lang[6], Stefan Janssen[13], Ute Fischer [1,2,14], Sanil Bhatia [1,2,14], Philipp A Lang[6], Arndt Borkhardt [1,2,14], Julia Hauer [7,15,16] & Aleksandra A Pandyra [1,2,3,4 ✉]

## Abstract

**Viral infections pose a significant global burden. Host susceptibility to pathogens is determined by many factors including genetic variation that can lead to immunodeficient or dysregulated antiviral immune responses. *Pax5* heterozygosity (*Pax5*−/+), resulting in reduced PAX5 levels in mice, mimics germline or somatic PAX5 dysregulation contributing to diseases such as childhood B-cell precursor acute lymphoblastic leukemia (B-ALL). In contrast to the well-characterized roles of PAX5 during early B-cell development, little is known about how *Pax5* heterozygosity impacts antiviral responses. We infected *Pax5*−/+ mice with the noncytopathic Lymphocytic Choriomeningitis Virus (LCMV) and found that infection with the chronic Docile strain resulted in decreased survival of *Pax5*−/+ mice. While early adaptive CD8+ T-cell (CTL) immunity was robust in *Pax5*−/+ mice, LCMV-specific neutralizing antibody production was compromised leading to impaired long-term viral clearance and a pro-inflammatory milieu in the bone marrow (BM). Here we show that survival outcomes were improved upon prophylactic treatment with the β-glucan immune trainer through induction of heterologous protection against chronic infection. β-Glucan enhanced viral clearance, CTL immunity, neutralizing antibody production and reduced monocyte immunosuppression in multiple LCMV-resident host organs. New insight from this study will help design effective prophylactic treatment strategies against chronic viral infections, particularly in genetically predisposed susceptible hosts.**

**Keywords** Trained Immunity; Chronic Infection; LCMV; PAX5; β-glucan
**Subject Categories** Cancer; Immunology; Microbiology, Virology & Host Pathogen Interaction

## Introduction

The transcription factor PAX5, being exclusively expressed in B lymphoid lineage cells, is a critical regulator of B-cell commitment (Fischer et al, 2020), development (Horcher et al, 2001; Nutt et al, 1999), and $V_HDJ_H$ recombination (Hill et al, 2020). PAX5 functions both as a transcriptional activator and repressor, the former of which ensures the expression of genes required for B-cell functional development (Schebesta et al, 2007) and the latter stymieing the expression of lineage-inappropriate genes (Revilla et al, 2012). These activating functions of PAX5 are closely linked to its role in leukemogenesis since PAX5 dysregulation often accompanies somatic loss of function mutations in leukemic cells or germline alterations leading to B-cell acute lymphoblastic leukemia (B-ALL) development in early childhood. *PAX5* genetic changes such as monoallelic deletions comprise the most common *PAX5* alterations (Familiades et al, 2009; Mullighan et al, 2007) and germline (R38H, G183S, and deletions) or somatic mutations (P80R and deletions)

[1]Department of Pediatric Oncology, Hematology and Clinical Immunology, Medical Faculty, Center of Child and Adolescent Health, Heinrich-Heine-University, Düsseldorf, Germany. [2]Center for Integrated Oncology Aachen Bonn Cologne Düsseldorf (CIO ABCD), Düsseldorf, Germany. [3]Institute of Clinical Chemistry and Clinical Pharmacology, University Hospital Bonn, Bonn, Germany. [4]German Center for Infection Research (DZIF), Partner Site Bonn-Cologne, Bonn, Germany. [5]Department of Molecular Medicine II, Medical Faculty and University Hospital, Heinrich-Heine-University, Düsseldorf, Germany. [6]Institute of Immunology, Medical Faculty, University of Duisburg-Essen, Essen, Germany. [7]Technical University of Munich, TUM School of Medicine and Health, Department of Pediatrics, Munich, Germany. [8]Institute of Pathology, Medical Faculty, Heinrich-Heine-University, Duesseldorf, Germany. [9]Department of Hematology, Oncology and Clinical Immunology, University Hospital Düsseldorf, Düsseldorf, Germany. [10]Molecular Medicine Partnership Unit, Heidelberg, Germany. [11]Spatial & Functional Screening Core Facility, Medical Faculty, Heinrich Heine University, Düsseldorf, Germany. [12]Faculty of Medicine, Tel-Aviv University, Tel Aviv, Israel. [13]Algorithmic Bioinformatics, Department of Biology and Chemistry, Justus Liebig University, Gießen, Germany. [14]DKTK partner site Essen-Düsseldorf, Düsseldorf, Germany. [15]German Cancer Consortium (DKTK), Partner Site Munich, a Partnership Between DKFZ and Technical University of Munich, Munich, Germany. [16]German Center for Child and Adolescent Health (DZKJ), Partner Site Munich, Munich, Germany. [17]These authors contributed equally: Zhe Lu, Olivia Stencel. ✉E-mail: apandyr1@uni-bonn.de

found in patients commonly result in reductions of PAX5 activity that mimic haploinsufficiency (Auer et al, 2014; Gu et al, 2019; Iacobucci et al, 2010; Shah et al, 2013). An increasing appreciation for the fact that these genetic lesions can function as predisposing B-ALL factors has been supported by accompanying in vivo work where *Pax5* haploinsufficiency combined with secondary stimuli/drivers initiate B-ALL. Specifically, *Pax5* heterozygous (*Pax5*$^{-/+}$) mouse models have been shown to cooperate with Early B-cell factor 1 (Ebf1) heterozygosity (Prasad et al, 2015), STAT5 activation (Heltemes-Harris et al, 2011) and non-specific facility-dependent infectious conditions (Martin-Lorenzo et al, 2015) to promote B-ALL. PAX5 fusions can cooperate with Janus Kinase 2 (JAK2) (Jurado et al, 2022), elastin (ELN) (Jamrog et al, 2018) and ETV6 (with loss of Cdkna2a/b) to initiate B-ALL (Smeenk et al, 2017). Taken together, *PAX5* genetic predisposing lesions resulting in haploinsufficiency are mimicked by the *Pax5*$^{-/+}$ mouse model that can cooperate with other genetic lesions and/or external stimuli to influence disease initiation and development.

Beyond their risk for developing B-ALL with a penetrance of about 50%, carriers of *PAX5* germline alterations exhibit mild signs of B-cell deficiency with significantly decreased numbers of memory and intermittent B cells foreseeably resulting from the block of B-cell differentiation at the pre-B1 stage (Escudero et al, 2022). Interestingly, no increased risk of infection or autoimmunity has been reported so far in carriers of germline *PAX5* heterozygous mutations. The effects on immunity in the context of *Pax5* heterozygosity (*Pax5*$^{-/+}$) are subtle and may not be apparent under naive steady-state conditions or following short-term antigen immunization challenge (Calderon et al, 2021; Kaiser et al, 2022; Smeenk et al, 2017). Furthermore, little is known about long-term responses to viral stimuli and accompanying changes in the bone marrow milieu in *Pax5*$^{-/+}$ hosts. In the current investigation, we employed two fundamentally different viruses, the acutely cytopathic Vesicular stomatitis Indiana virus (VSV) (Hangartner et al, 2006) and the noncytopathic Lymphocytic Choriomeningitis virus (LCMV) (Hangartner et al, 2006) to study the impact of *Pax5* heterozygosity on antiviral responses. The neurotropic VSV rabies-like virus must be rapidly cleared before it induces excessive damage in infected cells and is therefore known to elicit rapid neutralizing antibody responses. LCMV, in contrast, is a poorly cytopathic, persistent virus, whose clearance/control is dependent on T-cell activity. The LCMV-induced damage to the host is accordingly immune-related (immunopathology) and neutralizing antibody production is poor especially in the acute strains (Hangartner et al, 2006). In this study, we show that even though *Pax5*$^{-/+}$ did not hinder efficient viral clearance of the acute VSV infection, the long-term antiviral immune responses against LCMV chronic infection were severely affected. This culminated in a decreased ability to produce late-stage LCMV-neutralizing antibodies and subsequent failure to control chronic LCMV infection. We then demonstrated how susceptibility to chronic infection in the mildly immunocompromised host can be attenuated by prophylactic pre-treatment with β-glucan which improved antiviral immunity and decreased monocyte immunosuppression. β-glucan is known to induce trained immunity, a concept that describes the ability of innate immune cells such as monocytes to retain a memory-like phenotype driven by epigenetic programming following initial immune stimulation culminating in amplified responses upon secondary immune stimulation (Mitroulis et al, 2018).

Central immune training which originates in the bone marrow, and can last for months has been shown to influence the course of different pathologic conditions including responses to viral infections (Netea et al, 2023). Unlike some other well-known live-vaccine immune trainers, β-glucan is characterized by extremely favorable safety profiles and is easily administered (Geller and Yan, 2020). Importantly, the ability of β-glucan to reverse the deleterious effects of chronic infection has implications not only in the context of antiviral responses but also for B-ALL predisposed individuals.

## Results

### *Pax5* haploinsufficiency does not compromise CTL immunity but increases susceptibility to chronic viral infection

It was previously demonstrated that healthy carriers of heterozygous germline *PAX5* variants conferring susceptibility to B-ALL had a reduced percentage of intermittent and memory B cells compared to healthy controls (Escudero et al, 2022). When we evaluated other peripheral blood mononuclear cell (PBMC) lymphocyte populations in one of these families with a germline p.Gly183Ser mutation (Auer et al, 2014), naive CD4$^+$ and CD8$^+$ T-cell percentages were comparable to healthy reference controls (Comans-Bitter et al, 1997; Valiathan et al, 2014). However, memory T cells and NK cells were below the expected range (Fig. 1A). While PAX5 expression is limited to B cells, it is not unusual for B-cell abnormalities, especially those affecting B-cell development as reported in Common Variable Immunodeficiencies (CVI), to also affect the T-cell compartment (Fekrvand et al, 2022). This led us to investigate how hosts with *Pax5* germline alterations leading to mild B-cell deficiencies would respond to viral challenges.

Therefore, we decided to systemically challenge the mice with the cytopathic Vesicular Stomatitis Virus (VSV) whose control depends on the induction of interferons and very rapid early neutralizing antibody production (Hangartner et al, 2006) as well as activation of VSV-specific B cells in the marginal zone (MZ) (Junt et al, 2007). We reasoned that infection with a pathogen that is dependent on B-cell mediated immune responses could affect *Pax5*$^{-/+}$ mice differently than WT controls as has been previously demonstrated for hosts with severe B-cell defects that fail to mount sufficient anti-VSV immune responses (Honke et al, 2011; Khairnar et al, 2015). However, when *Pax5*$^{-/+}$ and WT mice were intravenously infected with VSV at a dose ($10^6$ PFU) previously used to discern differences between immunocompetent and their immunodeficient counterparts, *Pax5*$^{-/+}$ mice did not succumb to the infection (Fig. 1B). Early interferon production, a strong determinant of anti-VSV clearance, was not different between *Pax5*$^{-/+}$ and WT mice (Fig. EV1A). Given that VSV is largely neurotropic, replicates poorly in mice and is controlled very early (Hangartner et al, 2006), we surmised that the slight immunodeficiency incurred by *Pax5* haploinsufficiency was not enough to compromise responses following challenge with a rapidly cleared virus and where phenotypic differences might only arise in cases of severe compromising immunodeficiency in the B-cell compartment (Khairnar et al, 2015).

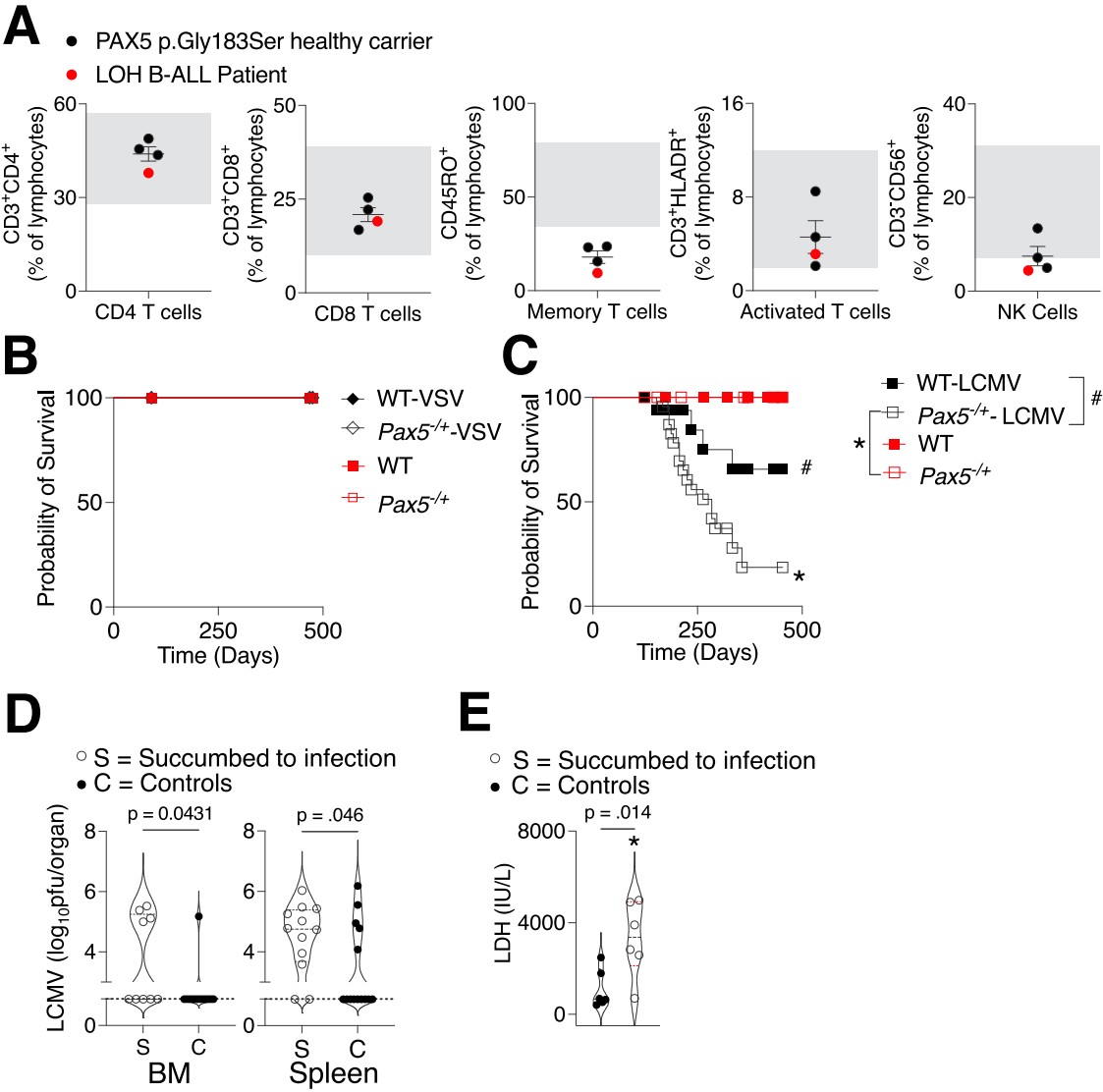

**Figure 1. *Pax5* haploinsufficiency confers an increased susceptibility to chronic but not acute viral infection.**

(A) Percentages of naive CD4$^+$ and CD8$^+$ T cells, memory and activated T cells, and NK cells in PBMCs of three healthy carriers and one patient (with *PAX5* loss of heterozygosity (LOH) in remission more than 5 years after completion of therapy) compared to reference healthy control ranges (gray box) are shown. (B) *Pax5$^{-/+}$* and WT mice were intravenously infected with 10$^6$ PFU of VSV and survival was monitored (*n* ≥10 mice per group). (C–E) *Pax5$^{-/+}$* and WT mice were intravenously infected with 10$^6$ PFU of LCMV Docile and (C) survival was monitored (*n* ≥12 mice per group) *$P$ = 0.0002, #$P$ = 0.0384. Statistical analyses were performed using Log-rank (Mantel–Cox) test with a Bonferroni correction for multiple curve comparisons. Where material was available, virus titers in the bone marrow (BM) and spleen (D) as well as LDH levels in the plasma (E, *n* = 6) were determined from mice in the survival curve in (C). S indicates that the mice were necessarily sacrificed as they succumbed to infection (*n* ≥9). C indicates healthy mice which were sacrificed as accompanying controls and/or at endpoint (*n* ≥13). Unless otherwise stated, statistical analyses were performed using a Student´s *t* test (unpaired, two-tailed). Error bars indicate SEM. Source data are available online for this figure.

Next, we investigated the challenge with a virus for which the mouse is a natural host. We therefore infected *Pax5$^{-/+}$* and WT mice with the noncytopathic Lymphocytic Choriomeningitis virus (LCMV) (Hangartner et al, 2006). There are several LCMV strains, acute and chronic, with a different range of effects on virus persistence, T-cell exhaustion and lethality (Suprunenko and Hofer, 2019; Takagi et al, 2017). Acute strains of LCMV are generally rapidly cleared and this was indeed the case for both *Pax5$^{-/+}$* and WT hosts when infected with 10$^6$ PFU of the Armstrong strain (Fig. EV1B,C). Therefore, we next infected mice with the chronic LCMV Docile strain (10$^6$ PFU) which is known to cause persistent

infection at higher doses (Khairnar et al, 2018; Ou et al, 2008). As with VSV, mice were observed for up to 16 months post infection. *Pax5$^{-/+}$* mice were more susceptible to the effects of the chronic infection than their WT counterparts (Fig. 1C). Mice were sacrificed due to weight loss and a moribund state attributable to an asymptomatic LCMV infection, with one mouse out of 23 developing T-cell acute lymphoblastic leukemia (T-ALL) at day 182 post infection (Fig. EV1D). Aside from the one case of T-ALL, which unthymectomized, genetically predisposed mice such as *Pax5$^{-/+}$* can be prone to when challenged with infectious and mutagenic agents (Dang et al, 2015; Schindler et al, 2009), there

were no B-ALL development in both infected or uninfected *Pax5*$^{-/+}$ animals unlike what was previously observed by others upon transfer of these mice from SPF to conventional (CF) conditions (Martin-Lorenzo et al, 2015). As all our experiments were carried out in the CF facility on a C57BL/6J background (backcrossed for at least ten generations), the lack of B-ALL development can likely be attributed to facility and strain-dependent differences, the latter of which were carried out on an unspecified percentage mix of C57BL/6 × CBA background (Martin-Lorenzo et al, 2015).

When we assessed the viral titers of *Pax5*$^{-/+}$ and WT mice at their endpoint or at sacrifice (from Fig. 1C) and respective healthy control mice, mice that were necessarily sacrificed had significantly higher viral titers than the respective controls in spleen and bone marrow (Fig. 1D) as well as elevated lactate dehydrogenase (LDH) levels which is consistently indicative of inflammation/tissue damage such as that resulting from chronic viral infection (Fig. 1E). Taken together, *Pax5*$^{-/+}$ were more susceptible to the effects of chronic infection and mice that succumbed the LCMV were characterized by detectable viral titer as well as elevated LDH levels.

When we assessed early cytotoxic T-cell (CTL) immune responses, we detected robust CD8$^{+}$ T-cell responses directed at H2-Db restricted tetramers (tet) against the LCMV glycoprotein (GP33–41, gp33$^{+}$tet$^{+}$CD8 T cell) and nucleoprotein (Db/NP396–404, np396$^{+}$tet$^{+}$CD8 T) epitopes in both *Pax5*$^{-/+}$ and WT mice in the liver, lymph node and bone marrow 15 days post infection (Fig. 2A). Tet$^{+}$CD8 T cells persisted steadily in the blood and were still present at day 120 post infection (Fig. 2B). There were no differences between viral titers in multiple organs at day 15 post infection (Fig. 2C). However, by day 120, while virus was largely cleared in the WT mice, *Pax5*$^{-/+}$ mice had detectable and significantly higher viral titers in the lymph nodes and spleen (Fig. 2C). Accordingly, this was accompanied by elevated levels of LDH in *Pax5*$^{-/+}$ mice 120 days post infection (Fig. 2D). When we examined hematoxylin and eosin (H&E) tissue sections, pericellular and perivascular inflammatory infiltrates were observed in both *Pax5*$^{+/-}$ and WT mice at 15 days post infection, which were largely absent in the WT mice but still persisted in *Pax5*$^{-/+}$ mice 120 days post infection (Fig. 2E). Taken together, while LCMV-specific CD8$^{+}$ CTL adaptive immune responses were elicited in both *Pax5*$^{-/+}$ and WT mice, the *Pax5* haploinsufficient hosts were slower at clearing the chronic LCMV which led to escape and persistent infection resulting in a significant increase in mortality in a greater proportion of mice.

## Neutralizing antibody production was severely compromised in *Pax5*$^{-/+}$ mice following chronic LCMV infection

Next, we wondered whether any mature B-cell perturbations were present in the spleen. Indeed, similar to what was observed in the PAX5G183S carriers, the lack of a *Pax5* copy caused decreased numbers of mature splenic B-cell populations particularly follicular B cells (FO), and marginal zone (MZ) B cells (Fig. 3A), both of which are important in mediating immune responses to pathogens (Pillai et al, 2004). Next, we checked non-specific immunoglobulin levels following chronic LCMV infection. While levels of IgM, IgG3, IgG2b, IgG2a, and IgG1 peaked at 30 days post infection, production of IgM and IgG2b was impaired in *Pax5*$^{-/+}$ mice (Fig. 3B). Although effective LCMV-specific neutralizing antibodies

do not develop in the early acute phase, chronic LCMV strains including Docile and Clone 13 have been shown to induce the production of neutralizing antibodies at later time points during infection (Ertuna et al, 2021; Eschli et al, 2007; Fallet et al, 2020). Indeed, when we checked the plasma of Docile-infected *Pax5*$^{-/+}$ and WT mice, detectable LCMV-neutralizing antibodies were present in the WT but not *Pax5*$^{-/+}$ mice 30 days post infection. Although neutralizing antibodies were also detected in the blood of *Pax5*$^{-/+}$ mice 90 days post infection, they were significantly lower than in the WT mice (Fig. 3C). The same was recapitulated when we tested for the presence of anti-LCMV GP-specific antibodies which were significantly lower in the plasma of *Pax5*$^{-/+}$-infected mice at day 90 and 120 post infection (Fig. 3D). Taken together, *Pax5*$^{-/+}$ mice produced dramatically lower LCMV-specific neutralizing antibodies post chronic LCMV infection leading to viral escape and persistent infection.

## *Pax5* haploinsufficiency shapes a distinct bone marrow microenvironment (BME) in response to chronic infection

The functional long-term consequences caused by *Pax5* haploinsufficiency originate from the bone marrow (BM). This is the case not only because of the immunodeficiency incurred by the B-cell differentiation block but also because chronic inflammation might affect somatic evolution (Li et al, 2023). Somatic evolution describes the accumulation of mutations over time that contribute to disease and ageing, and this is important in the context of preleukemic susceptibility to B-ALL. Furthermore, although the BM plays a critical role in the regulation of hematopoiesis, lineage-specific differentiation, as well as homing and survival of memory cells, the effects of viral persistence in the bone marrow microenvironment (BME) are relatively unexplored (Toppinen et al, 2021) but important not least because recent studies have detected the DNA of a plethora of viruses in the human BM (Toppinen et al, 2021). Therefore, we next sought to better characterize the BM of chronically infected *Pax5*$^{-/+}$ and WT mice.

We checked the levels of multiple cytokines using a LegendPlex™ panel in the BM and plasma at day 15 and 120 post infection. While IL-6, IL-10, IFN-α/β, TNF-α GM-CSF, and IL-12 were not changed or detectable in response to chronic LCMV infection compared to naive controls, there was a clear pattern of induction for other cytokines with some marked differences between *Pax5*$^{-/+}$ and WT (Figs. 4A and EV2). Specifically, there was a significantly stronger induction of IL-18, CCL3, CCL4, and CXCL1 in the BM of WT mice compared to *Pax5*$^{-/+}$ mice 15 days post infection. These cytokine increases in the BM of WT mice returned to steady-state levels at day 120 post infection. In the plasma of infected mice, IFN-γ, CXCL1, CCL4, CCL5 peaked at day 15 post infection in WT mice and then decreased to steady-state levels, whereas in *Pax5*$^{-/+}$ mice, CXCL10, CXCL9, CCL3, CCL4, and IFN-γ, remained elevated at day 120 post infection. Taken together, different patterns of cytokine induction were observed between *Pax5*$^{-/+}$ and WT mice following chronic infection with the former prone to a delayed response pointing to a longer-term pro-inflammatory state.

Next, we checked the antiviral CTL-mediated responses in the BM and found the persistence of both gp33$^{+}$tet$^{+}$CD8 T and np396$^{+}$tet$^{+}$CD8 T cells in the BM of *Pax5*$^{-/+}$ and WT chronically

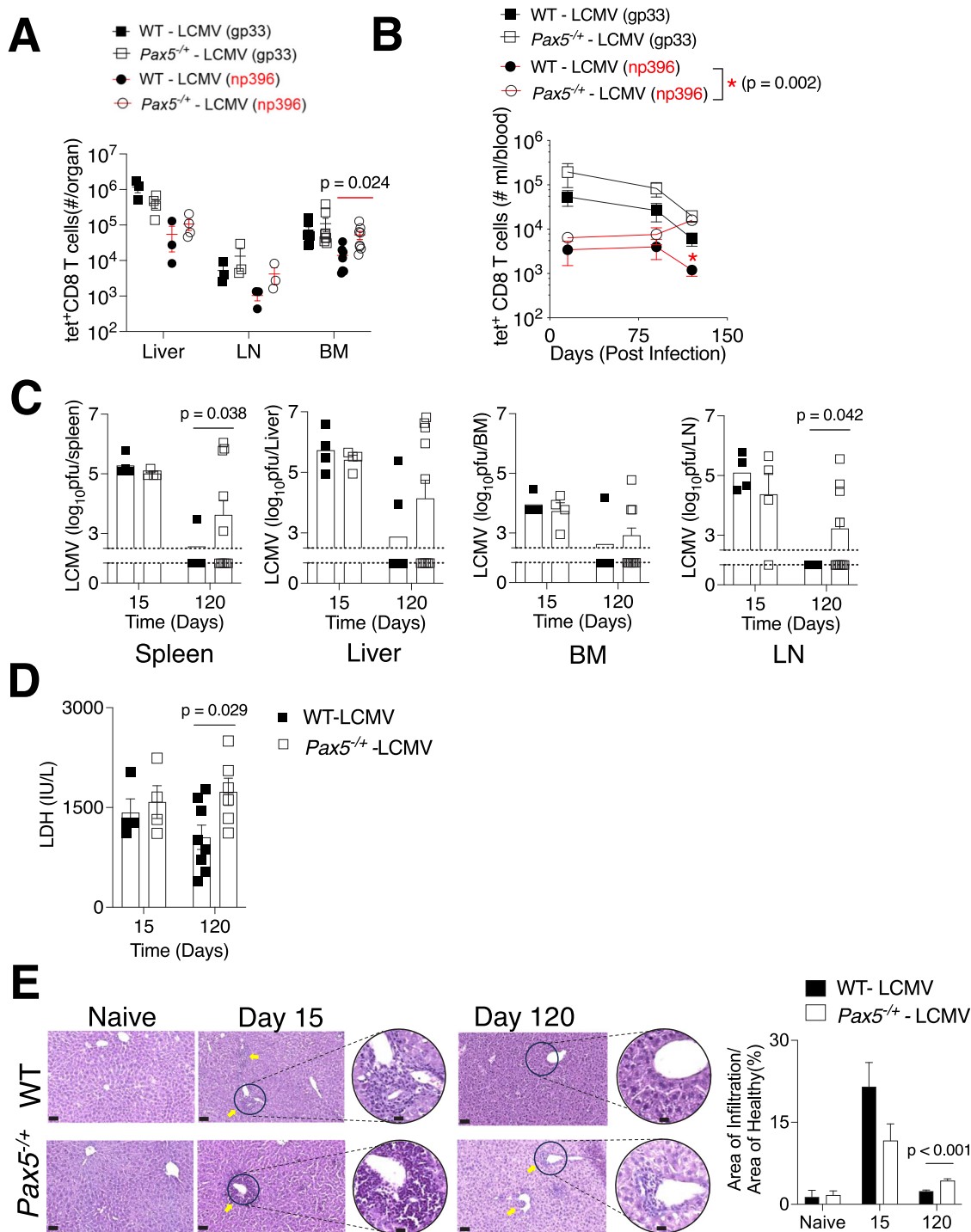

**Figure 2.** *Pax5* heterozygosity does not impair initial antiviral CTL responses but impacts long-term virus control.

(A–E) *Pax5*$^{-/+}$ and WT mice were intravenously infected with $10^6$ PFU of LCMV Docile and (A) Numbers of effector CTL tet-gp33$^+$ and tet-np396$^+$ CD8 T cells were measured at day 15 post virus inoculation in the liver, spleen lymph node (LN), and bone marrow (BM) using FACS (*n* ≥3 mice per group). (B) Effector tet-gp33$^+$ and tet-np396$^+$ CD8 T cells were measured in the blood at the indicated days post infection (*n* ≥3 mice per group). (C) LCMV virus titers were determined in the spleen, bone marrow (BM), lymph nodes (LN), and liver tissue at day 15 (*n* = 4) and 120 post infection using the plaque assay (*n* = 11). (D) LDH activity in the plasma was determined at day 15 (*n* = 4) and day 120 post infection (*n* ≥ 6). (E, left panel) Sections of snap-frozen liver tissues were analyzed using H&E staining (a representative set of images of *n* ≥3 mice per group is shown; scale bar = 50 μm; arrows indicate the inflammatory infiltrates). (E, right panel) Infiltrates were quantified. Error bars indicate SEM. Unless otherwise stated, all statistical analyses were performed by a Student´s *t* test (unpaired, two-tailed). Source data are available online for this figure.

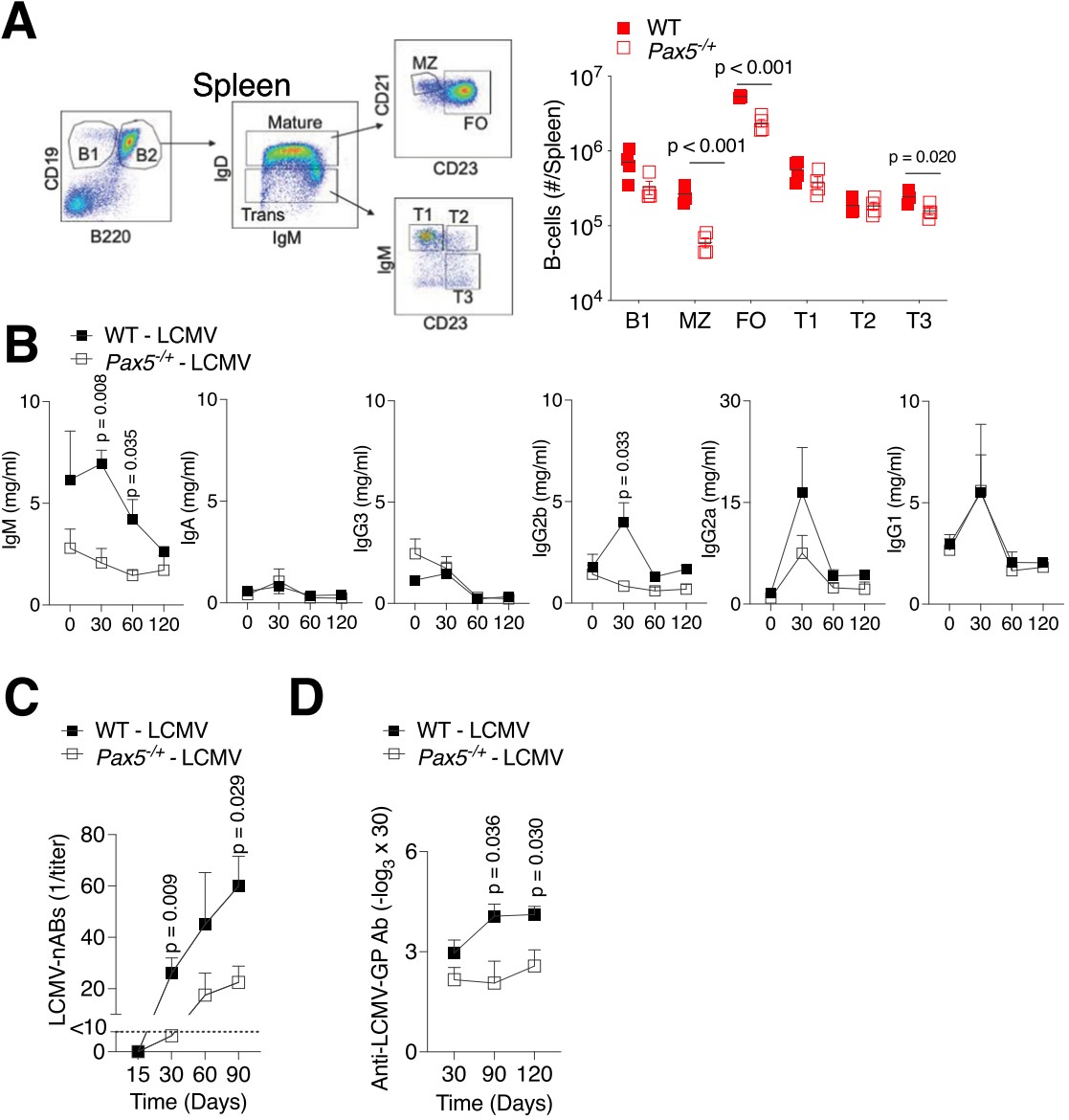

**Figure 3. Pax5 haploinsufficiency leads to impaired anti-LCMV neutralizing antibody production during chronic infection.**

(A) B-cell gating strategy for the assessment of mature splenic B cells is shown. Numbers of B1, marginal zone (MZ), follicular (FO), and transitional (T1, T2, and T3) B cells were assessed in the spleen of naive WT and $Pax5^{-/+}$ mice ($n = 4$ mice per group). (B–D) Immunoglobulins were evaluated at the indicated time points using the 6-PLEX Mouse Immunoglobulin Isotyping Panel ($n \geq 3$ mice per group). (C) LCMV-neutralizing antibodies in the plasma were indirectly assessed using plaque assay on the indicated days ($n \geq 4$ mice per group). (D) LCMV-specific neutralizing antibodies against the LCMV glycoprotein (GP) were assessed using ELISA on the indicated days, showing the fold increase from the background (naive serum) on the indicated days ($n = 4$ mice per group). Error bars indicate SEM. All statistical analyses were performed using a Student's $t$ test (unpaired, two-tailed). Source data are available online for this figure.

infected mice (Fig. 4B). While the frequencies of gp33+tet+CD8 T cells peaked at day 30 and were still present at day 120 post infection, the numbers of np396+tet+CD8 T cells also persisted in the BME and were significantly higher in the $Pax5^{-/+}$ mice at day 90 post infection (Fig. 4B). Furthermore, expression of exhaustion markers PD1, TIM-3 and LAG3 was elevated in $Pax5^{-/+}$ LCMV-specific CTL cells in the BM (Fig. 4C). Taken together, while the BM is an active site of expansion of LCMV-specific CD8+ T cells in both $Pax5^{-/+}$ and WT mice, LCMV-specific CTLs in the former are prone to exhaustion.

We then investigated the effects of chronic infection on B cells in the BM. As expected, given that at day 15 LCMV actively replicates in the BM of both $Pax5^{-/+}$ and WT mice, there was an initial depletion of Pro-B, Pre-BI, and pre-BII populations in both groups which then returned to homeostatic levels by day 30 post infection (Fig. EV3A). Next, we checked the expression of IL-7r and MHC-II on B-lineage precursor early subsets. IL-7r-mediated signaling drives the outgrowth of preleukemic clones to B-ALL (Clark et al, 2014; Geron et al, 2022). MHC-II has not only implications in immune recognition (Thibodeau et al, 2012) but

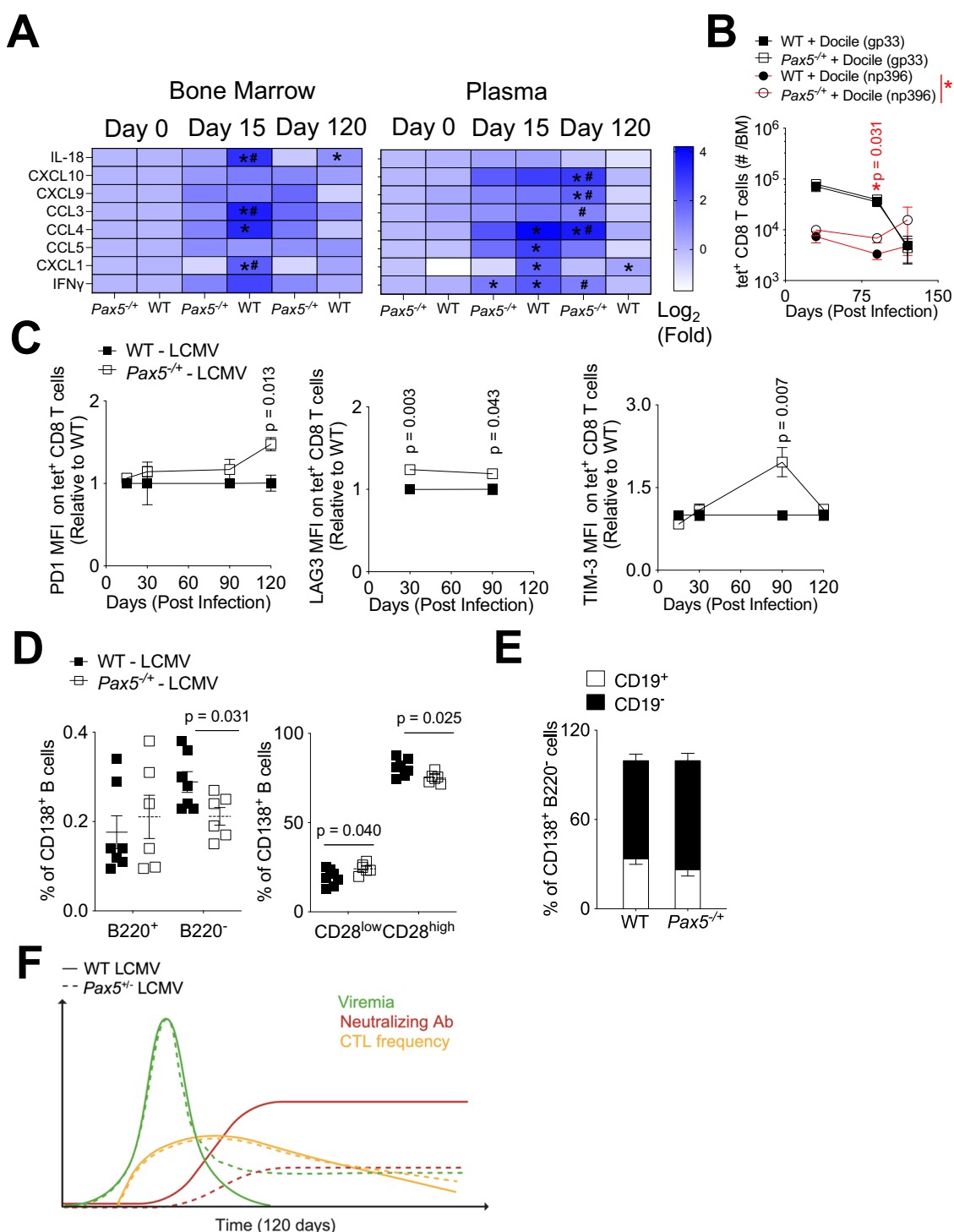

also has cell-intrinsic contribution in establishing the differentiated B-cell phenotype (Merkenschlager et al, 2019). To assess the long-term effects of infection, we chose the time point, 90 days post infection, where the frequencies of all subsets had stabilized in the context of the earlier acute-phase depletion stage. MHC-II expression was significantly increased in the pre-BI compartment in the BME of WT-infected but not Pax5$^{-/+}$-infected mice (Fig. EV3B). The pre-BII in both WT and Pax5$^{-/+}$-infected mice

had increased MHC-II expression but the upregulation was greater in the WT mice, while IL-7r expression remained largely unchanged (Fig. EV3B). We next checked plasma cells which are responsible for sustaining long-term antibody responses and can reside in the BM for many months (Slifka et al, 1998; Slifka et al, 1995). PAX5 has been shown to be expressed at relatively low levels in plasma cell subsets compared to other B cells in the bone marrow (Halliley et al, 2015). We corroborated this by extracting PAX5

**Figure 4. *Pax5* haploinsufficiency shapes a distinct bone marrow microenvironment (BME) in response to chronic infection.**

(A) Cytokine levels in the bone marrow and plasma were evaluated using the Procarta 19-Plex assay in uninfected naive controls (day 0), day 15 and day 120 post infection with $10^6$ PFU of LCMV Docile. Fold changes relative to day 0 genotype-specific controls are represented in a heatmap ($n = 3$ mice per group). *As indicated in the figure represents significant differences relative to time point 0 within each genotype as determined by a one-way ANOVA with a Dunnett's post hoc test. IL-18 (day 15 $P = <0.0001$, day 120 $P = 0.0471$), CXCL10 ($P = 0.0297$), CXCL9 ($P = 0.0467$), CCL3 ($P = 0.0003$), CCL4 (day 15 BM WT $P = 0.0016$, day 15 plasma WT $P < 0.0001$, day 120 plasma $Pax5^{-/+}$ ($P = 0.0401$), CCL5 ($P = 0.0016$), CXCL1 (BM $P = 0.0079$, plasma day 15 $P = 0.0002$, plasma day 120 $P = 0.0033$), IFN$\gamma$ ($Pax5^{-/+}$ $P = 0.0344$, WT $P = 0.0346$). #As indicated in the figure represents significant differences between the $Pax5^{-/+}$ and WT group at the indicated time point as determined by a Student's $t$ test (unpaired, two-tailed). IL-18 ($P = 0.038$), CXCL10 ($P = 0.009$), CXCL9 ($P = 0.006$), CCL3 (BM $P = 0.017$, plasma $P = 0.004$), CCL4 ($P = 0.020$), CXCL1 ($P = 0.01$), IFN$\gamma$ ($P = 0.037$). (B) $Pax5^{-/+}$ and WT mice were intravenously infected with $10^6$ pfu of LCMV Docile and effector CTL tet-gp33$^+$ and tet-np396$^+$ CD8 T cells were measured in the bone marrow at the indicated days post infection ($n \geq 3$ mice per group). (C) Surface molecule PD1, LAG3 and TIM-3 expression was measured on tet$^+$ CD8 T cells in the bone marrow at the indicated time points ($n \geq 3$ mice per group). (D) B220$^+$CD138$^+$ and B220$^-$CD138$^+$ cells were evaluated in the bone marrow as was the percent distribution of CD28 within the B220$^-$CD138$^+$ cell population 120 days post infection ($n \geq 6$ mice per group). (E) B220$^-$CD138$^+$ cells from (D) were further subdivided into percentages of CD19$^-$ and CD19$^+$ cells ($n \geq 2$ mice per group). Error bars indicate SEM. Unless otherwise stated, all statistical analyses were performed using a Student's $t$ test (unpaired, two-tailed). (F) The effects of chronic infection in $Pax5^{+/-}$ and WT mice are summarized (created in BioRender. Mescher M (2025) https://BioRender.com/b91v085). Source data are available online for this figure.

expression data from The Human Protein Atlas, Monaco dataset (Fig. EV3C). While the percent the of CD138$^+$B220$^+$ (plasmablasts) population was not different between $Pax5^{-/+}$ and WT mice 120 days post infection, WT mice had a higher percentage and number of CD138$^+$B220$^-$ cells (plasma cells) (Fig. 4D). The WT mice had a higher percentage of CD28$^+$ cells (Fig. 4D) which is known to maintain the metabolic fitness and survival of long-lived plasma cells (Utley et al, 2020) (Fig. 4D). Accordingly, while there were no differences in numbers of plasmablasts, plasma cell numbers were decreased in the BM of $Pax5^{-/+}$ mice at 120 days post infection (Fig. EV3D). We subdivided the plasma cells based on CD19 expression, CD19$^+$CD138$^+$B220$^-$ (plasma memory) cells and CD19$^-$CD138$^+$B220$^-$ (long-lived plasma memory) cells and there were no differences in distribution between $Pax5^{-/+}$ and WT hosts (Fig. 4E). In summary, the effects of chronic infection culminated in long-term inflammatory conditions in $Pax5^{-/+}$ hosts, increased CTL exhaustion, reduced plasma cells and decreased MHC-II expression on B-cell precursors cells. In the broad context, while early viremia and CTL responses were similar between WT and $Pax5^{-/+}$ hosts in response to chronic LCMV infection, neutralizing antibody production was compromised in the $Pax5^{-/+}$ hosts leading to exhaustion and virus persistence (Fig. 4F).

## Immune training ameliorates the effects of chronic LCMV infection, improves T-cell immunity, and affects immunosuppressive infiltrates

As chronic LCMV infection creates an inflammatory environment, we wondered about the possible implications of perturbing this state by training the immune system prior to infectious challenge. This, combined with previously published observations showing that inflammatory monocytes can hinder antiviral B-cell responses (Sammicheli et al, 2016), led us to pre-treat both $Pax5^{-/+}$ and WT mice with β-glucan, a well-known inducer of centrally trained immunity responses (Mitroulis et al, 2018). As expected, β-glucan treatment caused an expansion of LSK (Lin$^-$Sca-1$^+$c-KIT$^+$) cells in the BM of both $Pax5^{-/+}$ and WT mice (Figs. 5A and EV4A). Seven days post β-glucan treatment, we infected $Pax5^{-/+}$ and WT mice with LCMV (Fig. 5B). We found that β-glucan ameliorated the effect of chronic LCMV infection as all the WT mice survived till the observatory endpoint and a smaller proportion of $Pax5^{-/+}$ mice succumbed to the infection (Fig. 5C). When we evaluated viral titers at the endpoint and/or time of sacrifice in the case of the

$Pax5^{-/+}$ mice, there were no significant differences in viral titers unlike what we previously observed in Fig. 1C (Fig. 5D). Taken together, pre-treatment with the immune trainer β-glucan ameliorated the long-term deleterious effects of chronic infection (Fig. 5E).

In the context of antiviral immune responses, recent reports suggest that trained immunity elicits IFN-γ-dependent integrated organ immunity (Lee et al, 2024) and CX3CR1$^{hi}$ effector memory T cells (Tran et al, 2024). In the infected mice pre-treated with β-glucan, the virus was cleared faster and by day 30, a significantly higher proportion of mice had no detectable titers in the spleen and kidney (Figs. 6A and EV4B). In the lymph node and liver, only two out of seven mice had cleared the virus by day 30, while in the β-glucan pre-treatment group this was increased to five out of seven and four out of seven, respectively (Fig. EV4B). In line with these data, neutralizing antibody production was significantly improved at day 30 post infection in the β-glucan pre-treated group (Fig. 6B). As the potential immunomodulatory effects of β-glucan are not limited to the myeloid compartment (Ali et al, 2015), we evaluated the effects of β-glucan pre-treatment on the B-cell compartment in the bone marrow and spleen. As the most prominent effects in the BM B-cell compartment were evident 15 days post infection where we observed a depletion of pre-B subsets (pro-B, pre-BI, and pre-BII) in both $Pax5^{-/+}$ and WT mice (Fig. EV3A), we assessed pre-B-cell subsets at day 15 post LCMV infection in the presence of β-glucan. β-glucan pre-treatment did not increase the numbers of different B-cell subsets in the BM (Fig. 6C). However, β-glucan pre-treatment was able to ameliorate the depletion of pre-B-cell subsets LCMV-infected animals (Fig. 6C). β-glucan did not increase mature splenic B-cell numbers (Fig. EV4C). Furthermore, when we stained splenic sections of with Ki-67 and other immune markers using CO-Detection by indEXing (CODEX), there were no differences between the untreated and β-glucan-treated mice (Fig. EV4D).

When we checked CTL responses, LCMV-gp33$^+$tet$^+$ or LCMV-np396$^+$tet$^+$CD8 T-cell frequencies were increased at day 15 or day 30 in different organs harvested from β-glucan pre-treated LCMV-infected mice (Fig. 6D). Furthermore, when splenocytes were restimulated ex vivo with the LCMV-specific gp33 peptide, increased IFN-γ production was detected in CD8$^+$ T cells harvested from β-glucan pre-treated mice compared with infected controls at day 15 post infection (Fig. 6E). LDH levels in β-glucan pre-treated mice were also lower on day 15 post infection (Fig. 6F). Conversely,

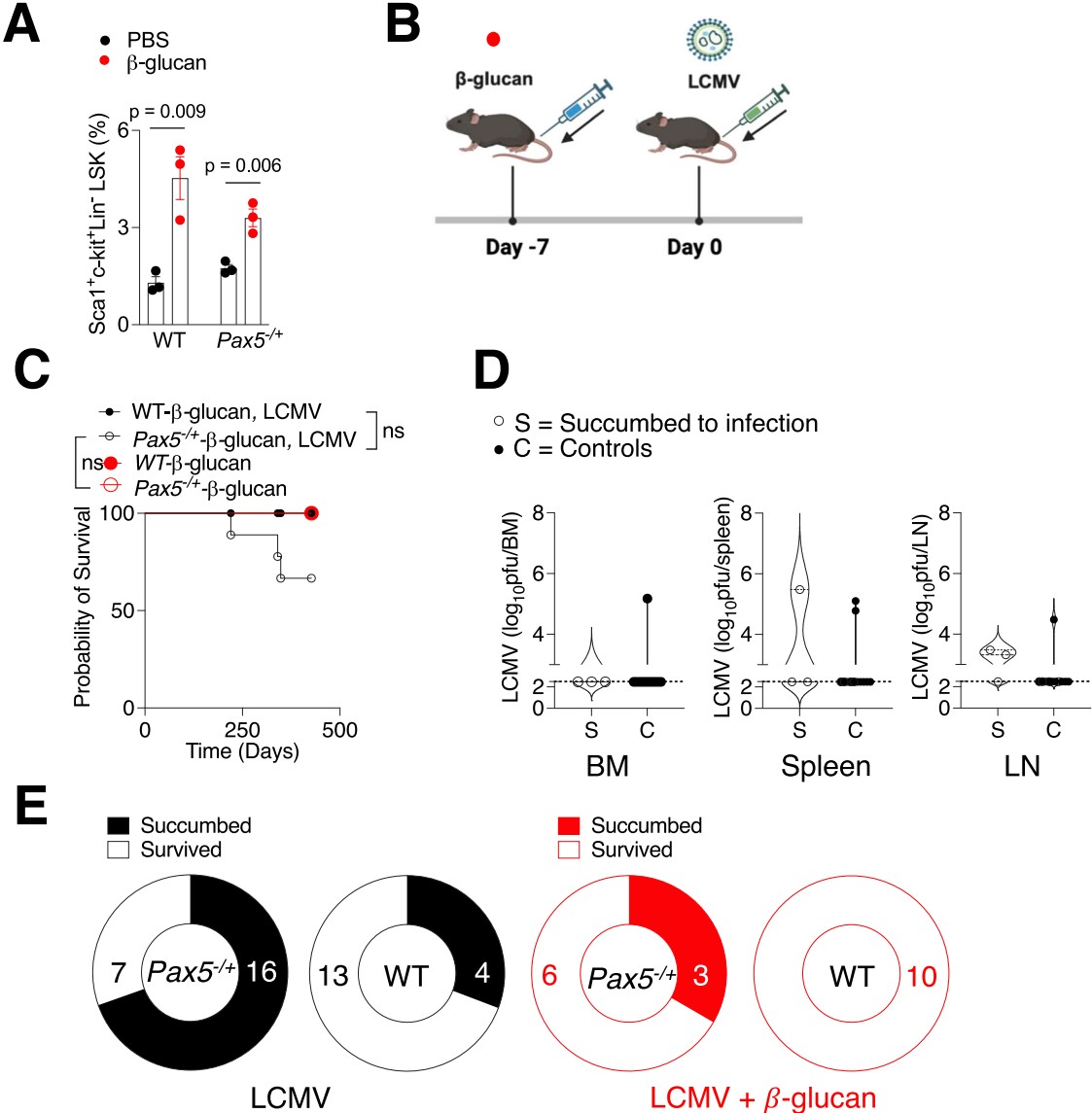

**Figure 5. Immune training with β-glucan improves outcomes following chronic LCMV infection.**

(A) WT and *Pax5⁻/⁺* mice were injected with 1 mg of β-glucan and LSK (Lin⁻Sca-1⁺c-KIT⁺) cells were measured in the bone marrow 24 h post treatment (*n* = 3 mice per group). (B) A schematic representation of the treatment regimen is shown (Created in BioRender. Mescher M (2025) https://BioRender.com/v92iO52). (C) *Pax5⁺/⁻* and WT mice were injected with 1 mg of β-glucan. Seven days later, this was followed by infection with 10⁶ PFU of LCMV Docile and survival was monitored (*n* ≥ 6 mice per group). Statistical analyses were performed using a Log-rank (Mantel–Cox) test with a Bonferroni correction for comparisons. (D) Virus titers were determined in the bone marrow (BM), spleen and lymph nodes (LN) of *Pax5⁻/⁺*, and WT mice from the survival curve in (C). S indicates that the mice were sacrificed before the endpoint and succumbed to infection (*n* = 3 mice). C indicates healthy mice which were sacrificed as accompanying controls and at endpoint (*n* ≥ 6 mice). (E) Survival proportions of LCMV Docile *Pax5⁻/⁺* and WT mice from Fig. 1B are directly compared to β-glucan pre-treated LCMV Docile-infected mice in (C). Error bars indicate SEM. Unless otherwise stated, all statistical analyses were performed using a Student´s *t* test (unpaired, two-tailed). Source data are available online for this figure.

there was a decreased infiltration of CD4⁺CD25⁺ regulatory T cells (Tregs) in the spleen at day 30 post infection in the β-glucan pre-treated mice (Fig. EV5A) and decreased FOXP3 expression of Tregs in the lymph nodes (Fig. EV5B). Myeloid cells including monocytes are known to be primary executers of immune training responses and as inflammatory monocytes have been shown to impair B cells antiviral immunity following LCMV infection (Sammicheli et al, 2016), we therefore evaluated monocyte infiltration in organs of infected and β-glucan pre-treated mice. Although

CD11b⁺Ly6G^low Ly6C^high monocyte infiltrate levels were the same in all tested organs (Fig. EV5C), monocytes in the liver, spleen, lymph nodes and bone marrow had significantly lower expression of the immunosuppressive PD-L1 marker in the β-glucan pre-treated infected mice (Fig. 6G). We next assessed multiple cytokines using the LegendPlex™ panel in the plasma at day 15 and 60 post infection in β-glucan pre-treated mice. Levels of pro-inflammatory cytokines CXCL9, CXCL10, as well as the inflammatory chemokines CCL3 and CCL5 were significantly lower at day

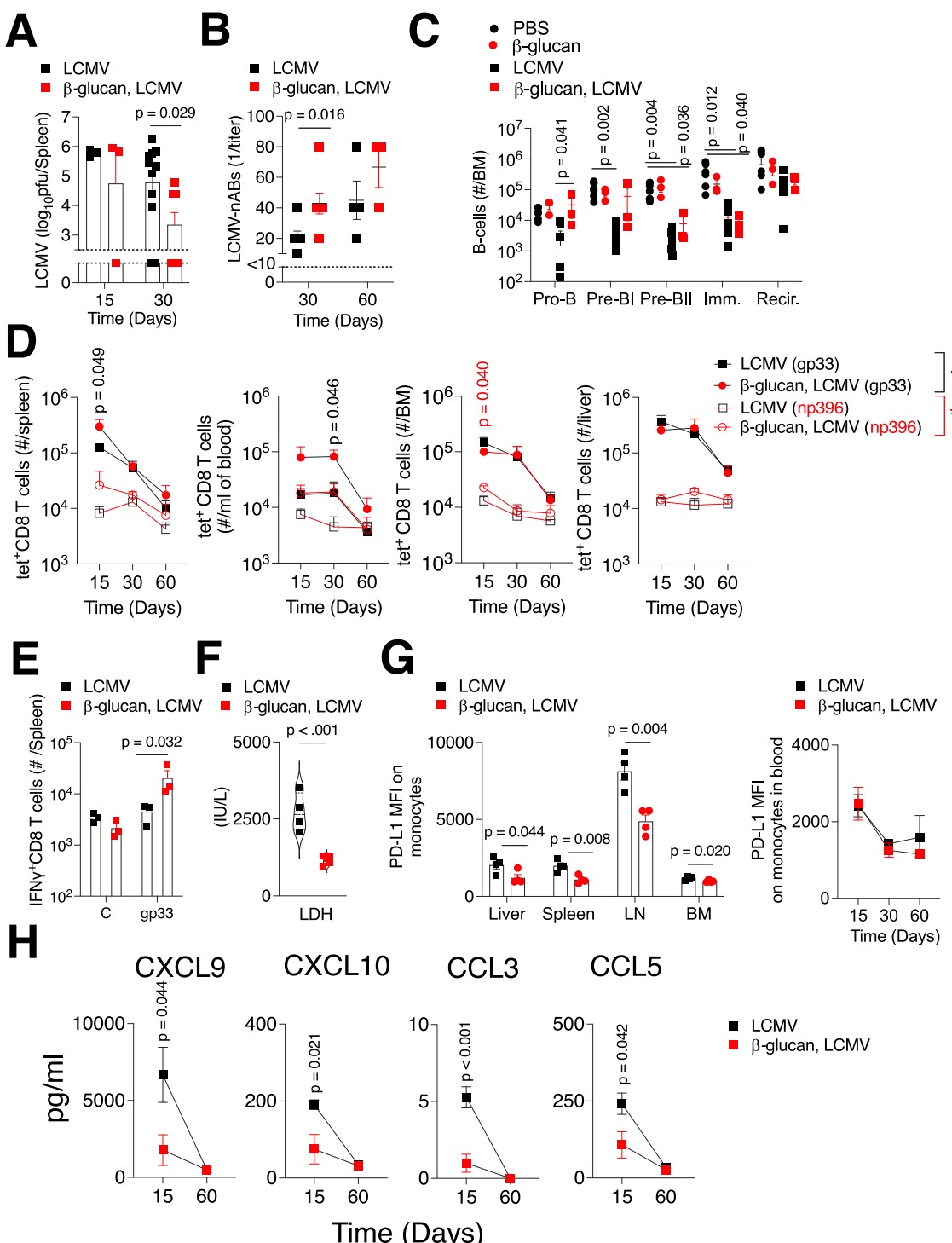

15 post infection in the β-glucan pre-treated group (Fig. 6H). Taken together, β-glucan pre-treatment induced protection against chronic LCMV infection by improving virus control and eliciting broad organ immunity including improved CTL and neutralizing antibody responses and decreased monocyte immunosuppression.

We wondered whether the administration of β-glucan would ameliorate the effects of chronic LCMV when administered post infection. We therefore reversed the treatments and applied β-glucan seven days post infection (Fig. 7A). LDH levels were lower in the β-glucan post infection treatment group compared to the infected controls 15 days post infection although this did not reach significance (Fig. 7B). However, the liver and spleen viral titers 30 days post infection were lower in the β-glucan-treated group (Fig. 7C). Furthermore, there were increases in splenic LCMV-

**Figure 6.  Immune training with β-glucan improves T-cell immunity and affects immunosuppressive infiltrates following chronic LCMV infection.**

(B–F) WT mice were injected with 1 mg of β-glucan and 7 days later this was followed by infection with $10^6$ PFU of LCMV Docile. (A) LCMV virus titers were determined in the spleen at day 15 ($n = 3$ mice per group) and day 30 post LCMV infection using plaque assay ($n \geq 7$ mice per group). (B) LCMV-neutralizing antibodies in the plasma were indirectly assessed using plaque assay on day 30 ($n = 7$ mice per group) and 60 post infection ($n \geq 3$ mice per group). (C) Pro-B, Pre-BI, Pre-BII, immature, and re-circulating B cells were measured in the bone marrow at day 15 post infection ($n \geq 3$ mice per group). Statistical analyses were performed using a one-way ANOVA with a Tukey post hoc test. (D) Numbers of effector CTL tet-gp33$^+$ and tet-np396$^+$ CD8 T cells were measured using FACS at the indicated time points in the blood, spleen, liver, and bone marrow ($n \geq 3$ mice per group). (E) 15 days post infection, single cell suspended splenic cells were restimulated with the gp33 LCMV-specific epitope, followed by intracellular staining for IFN-γ ($n = 3$ mice per group). Statistical analysis was performed using a one-way ANOVA with a Dunnett's post hoc test. (F) LDH activity in the plasma was determined 15 days post infection ($n \geq 4$ mice per group). (G) PD-L1 expression was determined on monocytes (CD11B$^+$Ly6G$^{low}$Ly6C$^{high}$) in the indicated organs ($n = 4$ mice per group) and blood ($n \geq 3$ mice per group) 30 days post infection. (H) Cytokine and chemokine levels in the plasma were evaluated using the Procarta 19-Plex assay in at day 15 post infection and significant differences between the β-glucan pre-treated infected and the infected alone group are shown ($n \geq 5$ mice per group). Error bars indicate SEM. Unless otherwise stated, all statistical analyses were performed using a Student's $t$ test (unpaired, two-tailed). Source data are available online for this figure.

gp33$^+$tet$^+$CD8 T cells 30 days post infection (Fig. 7D) but there were no differences in neutralizing antibody production (Fig. 7E). Taken together, although the administration of β-glucan post infection decreased viral titers, the effects were not as dramatic as observed with the β-glucan pre-administration. As by day 30 the re-stimulative capacity of CD8$^+$ T cells is diminished following infection with $10^6$ PFU of LCMV Docile, we used a lower dose of $10^4$ PFU (a dose at which the virus is cleared by day 30) to test the effects of pre and post of β-glucan treatment. Consistently, as with the higher LCMV dose, the monocytes in the LN had the highest PD-L1 expression which was significantly decreased in the β-glucan pre-treated group (Fig. 7F). PD-L1 expression remained comparatively low in the other organs although it was significantly lower in the blood monocytes in both β-glucan pre-and post-treatment groups (Fig. 7F). Upon ex vivo restimulation with the gp33 peptide, granzyme B expression was significantly higher in CD8$^+$ T cells of the lymph node and liver in the β-glucan pre-treated group (Fig. 7G). KLRG1, expressed on short lived effector cells (Xu et al, 2021) was elevated in CD8$^+$ T cells in the spleen and liver of β-glucan pre-treated mice and also in the liver of β-glucan post-treated mice at 30 days post infection (Fig. 7H). Consistently, there was a higher percentage of LCMV-gp33$^+$tet$^+$CD8 T cells in the organs of β-glucan pre-treated mice (Fig. 7I). At a dose $10^4$ LCMV Docile PFU, the structure of the lymphoid organs is intact 30 post infection (Fig. 7J) and co-staining with Ly6C (monocytes), CD3 (T cells) and B200 (B cells) revealed areas of proximity between monocytes-T cells and monocytes-B cells in all treatment groups (Fig. 7J) suggesting that monocytes in the tissue microenvironment can potentially exert immunomodulating functions such as for example through the PD-L1–PD1 axis.

## Discussion

Although specific pathogen susceptibilities are well documented in children and adults with genetically classifiable primary immuno-deficiencies (Huck et al, 2009; Redmond et al, 2022), less is known about other genetic predispositions and how clinically related they are to immune dysregulation. While loss of function germline and somatic alternations in *PAX5* predispose children to B-ALL, our study raises the question of whether these individuals are more likely to be heavily burdened following chronic viral infection. Carriers of *PAX5* germline mutations had less intermittent and memory B cells (Escudero et al, 2022), a feature also present in children with common variable immunodeficiencies (Piatosa et al,

2013). Furthermore, the stress of a persistent inflammation and viremia associated with a chronic infection in slightly immunode-ficient hosts could potentially trigger a secondary hit necessary for B-ALL development as was previously shown using in vivo models where *Pax5*$^{-/+}$ mice in a mixed C57BL/6 × CBA more tumor-susceptible background (Rithidech et al, 1999) were transferred to a conventional facility (CF) (Martin-Lorenzo et al, 2015). In our backcrossed C57BL/6J models, uninfected *Pax5*$^{-/+}$ mice housed in CF conditions did not develop leukemia within the 16-month observational span. The chronically infected *Pax5*$^{-/+}$ mice on the other hand succumbed to infection likely preceding any potential leukemia development although none was observed following VSV and acute LCMV infection.

In previous studies, B-cell frequencies in the BM and spleen were largely unchanged in *Pax5*$^{-/+}$ mice (Smeenk et al, 2017) and reconstitution following competitive bone marrow transplantation was unaffected (Smeenk et al, 2017). Mice with bi-allelic Pax5 (R31Q, *Pax5*$^{R31Q/-}$) mutations but not *Pax5*$^{-/+}$ mice, had hypo-gammaglobulinemia and disturbances in B-cell numbers in the BM, blood and spleen (Kaiser et al, 2022). Accordingly, this was accompanied by ablation of B-cell immune responses in *Pax5*$^{R31Q/-}$ mice following immunization with the immunogenic protein NP-KLH. Although the short-term, 14-day production of antigen-specific IgG and IgG1 antibodies was not impaired in *Pax5*$^{-/+}$ mice, the numbers of splenic GC B cells were slightly decreased relative to *Pax5*$^{+/+}$ controls (Kaiser et al, 2022). Our results recapitulate these observations as challenge with VSV, a neurotropic virus that is typically rapidly cleared, resulted in neutralizing antibody production in both *Pax5*$^{-/+}$ and WT mice.

However, immunization with a persistent chronic infection uniquely challenges the immune system. Responses following chronic infection can be chiefly skewed towards promoting efficient germinal B center responses even in viruses relatively independent from B cells such as LCMV (Fallet et al, 2020). Although rapidly cleared acute LCMV infection generally fails to elicit neutralizing antibody responses (Eschli et al, 2007) chronic LCMV infection not only drives functionally productive and protective GC B-cell responses, but also elicits plasma cell and memory B-cell output from GCs (Fallet et al, 2020). Therefore, under homeostatic circumstances and in cases of acute infection or short-term immunization (Kaiser et al, 2022), the subtle immunodeficiency incurred by the *Pax5* heterozygosity might not have functional consequences. However, differences become apparent during chronic infection which can further dysregulate humoral immunity

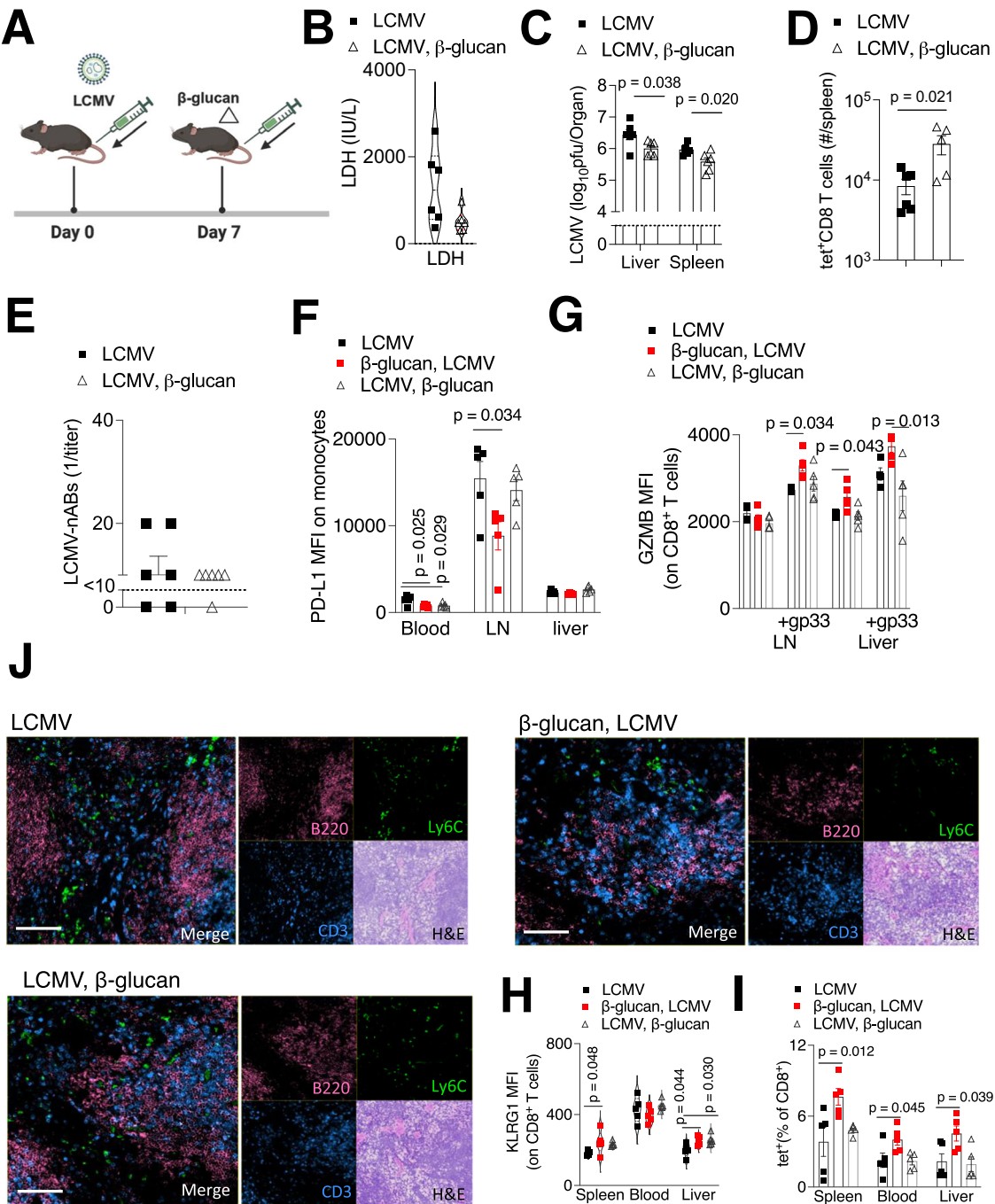

as is also observed with HIV, hepatitis C virus (HCV) and chronic hepatitis B viruses (Cooper and Good-Jacobson, 2020). Thus, despite early CTL responses being uncompromised, the inability to effectively mount neutralizing antibodies response resulted in virus persistence Pax5⁻/⁺ mice. It is conceivable that the compromised neutralizing antibody production and decreased BM plasma cells in Pax5⁻/⁺ hosts intrinsically stems from the skewered decreased ratio of mature B/early progenitor BM B cells. While short-term immunization and acute infections are easily compensated for, in the context of a chronic infection the situation is further exacerbated by persistent inflammation in the bone marrow and

systemic circulation. In contrast to the WT hosts, where many pro-inflammatory cytokine and chemokine levels were elevated at day 15 post infection in the plasma but returned to normal levels by day 120, CXCL9-10, and CCL3-5 persisted in the plasma of Pax5⁻/⁺ mice at day 120 post infection. As persistent inflammation can detrimentally affect antiviral LCMV B cells (Fallet et al, 2016), this likely furthermore contributed to the already compromised neutralizing antibody production and T-cell exhaustion in Pax5⁻/⁺ mice.

Most notably, pre-treatment with β-glucan improved survival outcomes in LCMV-infected Pax5⁻/⁺ and WT mice. β-glucan is known

**Figure 7. β-glucan maintains some protective effects against chronic LCMV when administered post infection.**

(A) WT mice were infected with $10^6$ PFU of LCMV Docile and 7 days later treated with 1 mg of β-glucan. Created in BioRender. Mescher M (2025) https://BioRender.com/p95y241 (B) LDH activity in the plasma was determined 15 days post infection ($n = 6$ mice per group). (C) LCMV virus titers were determined in the spleen and liver at day 30 post LCMV infection using plaque assay ($n \geq 5$ mice per group). (D) Number of effector CTL tet-gp33$^+$ T cells were determined in the spleen using FACS analysis 30 days post infection ($n = 5$–6). (E) LCMV-neutralizing antibodies in the plasma were indirectly assessed using plaque assay on day 30 post infection ($n = 6$ mice per group). (F–H) WT mice were injected with 1 mg of β-glucan, and 7 days later this was followed by infection with $10^4$ PFU of LCMV Docile. One infected-only group was also treated with 1 mg of β-glucan 7 days post infection. (F) PD-L1 expression was determined on monocytes in the indicated organs 30 days post infection ($n = 5$ mice per group). Statistical analyses were performed using a one-way ANOVA with a Tukey post hoc test. (G) 30 days post infection, singly suspended cells from the lymph node (LN) or liver were restimulated with the gp33 LCMV-specific epitope, followed by intracellular staining for granzyme B expression (GZMB) ($n \geq 4$ mice per group). Statistical analyses (F, G) were performed using a one-way ANOVA with a Tukey post hoc test. (H) Expression of KLRG1 on CD8$^+$ T cells in the spleen, blood, and liver was assessed 30 days post infection ($n = 5$ mice per group). (I) Effector CTL tet-gp33$^+$ T cells (expressed as percent of CD8$^+$ T cells) were assessed in the spleen, blood and liver 30 days post infection ($n = 5$). Statistical analyses (H, I) were performed using a one-way ANOVA with a Dunnett's post hoc test (J) Splenic sections were stained with the indicated antibodies and detected using CO-Detection by indEXing (CODEX), scale bar indicates 50 µm; (a representative image of $n$ of 5 (mice per group) is shown). Error bars indicate SEM. Unless otherwise stated, statistical analyses were performed using a Student´s $t$ test (unpaired, two-tailed). Source data are available online for this figure.

to induce immune training and increase immune responses upon secondary challenge (Kalafati et al, 2020; Mitroulis et al, 2018; Ochando et al, 2023; Ziogas et al, 2023). In the context of diseases such as infections or autoimmunity, these hyper-responsive effects can be protective or detrimental respectively. In the case of β-glucan, supplements have been demonstrated to decrease lower respiratory tract infections in children (Jesenak et al, 2013) and common colds in adults (Auinger et al, 2013), but these studies did not differentiate the general immunostimulatory from prophylactic immune training effects. While recently immune training and subsequent cross-protective antiviral immunity have been demonstrated with the commonly administered Bacillus Calmette–Guerin (BCG) vaccine against influenza (Tran et al, 2024) and SARS-CoV-2 (Lee et al, 2024), we chose to use β-glucan, deeming it a safer alternative for administration to immunocompromised hosts as BCG is a live vaccine.

A single β-glucan dose prior to chronic LCMV infection improved T-cell immunity and resulted in significant virus clearance by day 30 following chronic LCMV infection. The β-glucan-mediated effects were related to improved multi-organ immunity affecting several immune subsets and responses related to LCMV control. Although β-glucan administration alone did not induce proliferation of mature splenic B-cell subsets and immature and pre/pro-B-cell subsets, it did partially alleviate the BM depletion induced by chronic infection 15 days post infection. This is corroborated by others who have shown that in ex vivo systems although β-glucan failed to induce B-cell proliferation and IgM production, it increased B lymphocyte activation (Ali et al, 2015). CD11b$^+$Ly6C$^{hi}$ inflammatory monocytes have been demonstrated to suppress anti-LCMV B-cell responses (Sammicheli et al, 2016) and β-glucan pre-treatment decreased immunosuppressive PD-L1 (Cane et al, 2019) expression in monocytes at day 30 post LCMV infection, at a time point when neutralizing antibody production was commencing in our system and indeed was increased by β-glucan pre-treatment. The improvement in CTL immunity by day 15 as evidenced by increased numbers of LCMV-specific CD8$^+$ T cells as well as decreased LDH levels preceded neutralizing antibody production and viral titers clearance suggesting that the latter are a consequence rather than cause of the overall improvement in CTL immunity. The increase in CTL fitness was further corroborated by increased granzyme B expression following restimulation with the LCMV-specific gp33 peptide following infection with lower Docile LCMV doses. This was particularly relevant in the LN where the most consistent

reduction in monocyte PD-L1 expression occurred, suggesting a direct de-repression of CTL function through the PD1/PD-L1 axis. It can be further speculated that the reduction in pro-inflammatory chemokines and cytokines observed with the β-glucan pre-treatment at day 15 post infection alleviated immunosuppression mediated by chronic inflammatory signals in both monocytes and CTLs. Although there were some improvements in viral titers and CTL immunity when β-glucan was administered post infection, these are likely uncoupled from the contribution stemming from the CD11b$^+$Ly6G$^{low}$Ly6C$^{hi}$ inflammatory monocytes since minor differences in PD-L1 expression were observed.

This ability of β-glucan to ameliorate the effects of chronic infection has potential implications for B-ALL genetically predisposed children especially since they are routinely administered vaccines. This could impact vaccine recommendations (i.e., live vs. non-live) and between known immune trainers (such as BCG and MMR) (Hauer et al, 2021; Ziogas et al, 2023) and those that are not. Although it has been suggested that early vaccinations in children incur protective effects (Hauer et al, 2020; Morra et al, 2017; Singh et al, 2023), controlled and prospective studies are needed to further delineate the potential relationship between early immune training and heterologous antiviral protection. Furthermore, aside from the prophylactic training angle, more studies are needed to fully explore and optimize the potential of β-glucan as a therapeutic antiviral and to delineate the contribution of training vs. interventional treatment in the specific context of antiviral B- and T-cell immune responses.

## Methods

**Reagents and tools table**

| Reagent/resource | Reference or source | Identifier or catalog number |
|---|---|---|
| **Experimental models** | | |
| C57BL/6J mice | Charles River | N/A |
| L929 cells (*M. musculus*) | ATCC | CCL-1 |
| LCMV Armstrong | Rolf Zinkernagel, University of Zurich, Zurich, Switzerland | N/A |

| Reagent/resource | Reference or source | Identifier or catalog number |
|---|---|---|
| LCMV Docile | Dr. C. J. Pfau, Troy, NY | N/A |
| MC57G cells (*M. musculus*) | ATCC | CRL-2295 |
| *Pax5*$^{-/+}$ heterozygous mice | ZETT, Heinrich-Heine-University Düsseldorf, Germany | |
| *Pax5*$^{+/+}$ wildtype mice | ZETT, Heinrich-Heine-University Düsseldorf, Germany | |
| VSV Indiana strain, Mudd-Summers isolate | Professor D. Kolakofsky (University of Geneva, Switzerland) | |
| **Recombinant DNA** | | |
| **Antibodies** | | |
| Anti-Human Fc | Jackson ImmunoResearch | 109-001-008 |
| B220 | BD Biosciences | 557390 |
| B220, Alexa Fluor 700 | Thermo Fisher Scientific | 56-0452-82 |
| B220, PerCP-Cy5.5 | Thermo Fisher Scientific | 45-0452-80 |
| CD117, FITC | Thermo Fisher Scientific | 11-5998-82 |
| CD117, PE-Cy7 | BD Biosciences | 561681 |
| CD11b, BV605 | BD Biosciences | 567202 |
| CD127 (IL7R), FITC | Thermo Fisher Scientific | 11-1271-82 |
| CD127 (IL7R), PercP-Cy5.5 | Thermo Fisher Scientific | 45-1271-82 |
| CD127, Alexa Fluor 700 | Thermo Fisher Scientific | 56-1271-82 |
| CD138, APC | BD Biosciences | 558626 |
| CD138, PE | Thermo Fisher Scientific | MA5-23527 |
| CD150, APC | Invitrogen | 17-1502-80 |
| CD19, BV786 | BD Biosciences | 563333 |
| CD21, APC-Cy7 | BioLegend | 123418 |
| CD21, FITC | Thermo Fisher Scientific | 11-0211-82 |
| CD23, BV650 | BD Biosciences | 740591 |
| CD25, PE | Miltenyi Biotec | 130-120-766 |
| CD25, PE-Cy7 | Thermo Fisher Scientific | 25-0251-82 |
| CD25/IL-2r | R&D systems | AF2438 |
| CD28, PE | Thermo Fisher Scientific | 12-0289-42 |
| CD3 | BD Biosciences | 555273 |
| CD3, APC-eFluor 780 | Thermo Fisher Scientific | 47-0031-92 |
| CD3, PE-Violet770 | Miltenyi Biotec | 130-116-490 |
| CD4 | BD Biosciences | 553043 |
| CD4, BV450 | BD Biosciences | 560468 |
| CD4, SuperBright 600 | Thermo Fisher Scientific | 63-0042-82 |
| CD48, SuperBright 600 | Invitrogen | 63-0481-80 |
| CD62L, Alexa Fluor 700 | Thermo Fisher Scientific | 56-0621-82 |
| CD8a, BV510 | BD Biosciences | 563068 |

| Reagent/resource | Reference or source | Identifier or catalog number |
|---|---|---|
| CD8a, BV711 | Thermo Fisher Scientific | 407-0081-82 |
| CD8a, PerCP-Cy5.5 | Thermo Fisher Scientific | 45-0088-42 |
| DAPI | Thermo Fisher Scientific | 48-5993-82 |
| ECL rabbit IgG, HRP-linked | Cytiva, Amersham Biosciences | NA934 |
| FOXP3 | Thermo Fisher Scientific | 14-5773-82 |
| FOXP3, PerCP-Cy5.5 | Thermo Fisher Scientific | 45-5773-82 |
| Granzyme B, PE | Thermo Fisher Scientific | 12-8898-82 |
| IFN-γ, APC | Thermo Fisher Scientific | 17-7311-82 |
| HRP-labeled anti-Mouse IgG | Sigma | A3673 |
| IgD, eFluor 450 | Thermo Fisher Scientific | 48-5993-82 |
| IgD, SuperBright645 | Thermo Fisher Scientific | 64-5993-82 |
| IgM, FITC | Thermo Fisher Scientific | 11-5790-81 |
| IgM, PE-Cy7 | BD Biosciences | 552867 |
| Ki-67 | BD Biosciences | 556003 |
| KLRG1, BV650 | BD Biosciences | 740553 |
| LAG3, BV711 | Thermo Fisher Scientific | 407-2239-42 |
| LCMV Gp-Fc | In-house | N/A |
| Ly6C | Novus Biologicals | NBP1-28046 |
| Ly6C, Alexa Fluor 700 | BD Biosciences | 561237 |
| Ly6G, FITC | Thermo Fisher Scientific | 11-9668-82 |
| MHC-II, APC-eFluor 780 | Thermo Fisher Scientific | 47-5321-82 |
| Mouse Hematopoietic Lineage Antibody Cocktail, FITC | Thermo Fisher Scientific | 22-7770-72 |
| PD1, eFluor 610 | Thermo Fisher Scientific | 61-9985-82 |
| PD-L1, PE | BioLegend | 155404 |
| Sca-1, PE | BD Biosciences | 553108 |
| TIM-3, PE | Thermo Fisher Scientific | 12-5870-82 |
| VL-4 rat anti-LCMV NP mAB | In-house | N/A |
| **Chemicals, enzymes, and other reagents** | | |
| β-glucan | InvivoGen | tlrl-bgp |
| Acetone | Supelco | 1.00658 |
| Alpha MEM Eagle | PAN-Biotech | P04-21502 |
| Citric acid | Honeywell-Fluka | 15662880 |
| CODEX staining kit | Akoya Biosciences | SKU 7000008 |
| DMEM | Thermo Fisher Scientific | 31966-021 |
| DPBS | Sigma-Aldrich | D8537 |
| Eosin Y | Sigma-Aldrich | 318906 |
| FACS Lysing Solution 10X Concentrate | BD Biosciences | 349202 |
| Fetal bovine serum (FBS) | Sigma-Aldrich | F9665 |
| Fixable Viability Dye eFluor™ 520 | Thermo Fisher Scientific | 65-0867-14 |

| Reagent/resource | Reference or source | Identifier or catalog number |
|---|---|---|
| Formaldehyde 37% | Sigma-Aldrich | 1040021000 |
| $H_2O_2$ | Sigma-Aldrich | 21.6676-3 |
| Haematoxylin | Sigma-Aldrich | H3136 |
| Hanks' Balanced Salt Solution | PAN-Biotech | P04-34500 |
| IMDM | Thermo Fisher Scientific | 12440-053 |
| Methylcellulose | Honeywell-Fluka | 64620 |
| $Na_2HPO_4 \cdot 2H_2O$ | Honeywell-Fluka | FLK.30435.1000.P |
| Penicillin/ Streptomycin | Sigma-Aldrich | P4333 |
| RBC lysis buffer | pluriSelect | 60-00050-11 |
| Ortho-Phenylendiamin | Sigma-Aldrich | P3888 |
| Tissue-Tek OCT Compound | Sakura Finetek USA | SA62550-01 |
| Triton X-100 | Sigma-Aldrich | X-100 |
| 0.05% Trypsin/EDTA | Thermo Fisher Scientific | 25300-054 |
| **Oligonucleotides and other sequence-based reagents** | | |
| Oligonucleotides for CODEX conjugation | Dr. Peter-Martin Bruch, University Hospital Düsseldorf, Düsseldorf, Germany | See Table EV1 |
| GP33 peptide | Thermo Fisher Scientific | N/A |
| H-2-D(b) LCMV GP var c41M 33-41, KAVYNFATM, biotinylated monomer | NIH Tetramer Core Facility | 119290 |
| H-2-D(b) LCMV NP396-404, FQPQNGQFI, biotinylated monomer | NIH Tetramer Core Facility | 119291 |
| **Software** | | |
| FlowJo v10.8.1 | | https:// www.flowjo.com/ solutions/flowjo/ downloads |
| GraphPad Prism v10 | | https:// www.graphpad.com |
| GraphPad Prism v9 | | https:// www.graphpad.com |
| ImageJ | | https://imagej.net/ij/ |
| QuPath-0.5.0 | | https://qupath.github.io |
| QuPath-0.5.1 | | https://qupath.github.io |
| **Other** | | |
| Aperio slide scanner | Leica, Nussloch, Germany | |
| CountBright™ Absolute Counting Beads | Thermo Fisher Scientific | C36950 |
| Cell culture microplate, 24-well, transparent | Greiner | 662160 |
| Cell culture microplate, 96-well, transparent | Greiner | 650188 |
| CytoFlex | Beckmann Coulter | |

| Reagent/resource | Reference or source | Identifier or catalog number |
|---|---|---|
| Fortessa | BD Biosciences | |
| Drierite™ beads | Thermo Fisher Scientific | 219065000 |
| Foxp3/Transcription Factor Staining Buffer Set | Thermo Fisher Scientific | 00-5523-00 |
| Histosette I | Simport | M491-4 |
| LED panel, 4100-4600LUX | AGPTEK | |
| Mouse Immunoglobulin Isotyping Panel (6-plex) | BioLegend | 740492 |
| PhenoCycler Fusion | Akoya Biosciences | |
| Mouse ProcartaPlex Mix&Match 19-plex | Thermo Fisher Scientific | PPX-19-MXDJZDN |
| Spotchem EZ SP-4430 | ARKRAY | |
| Spotchem II Liver-1 strips | ARKRAY | 09636307001 |
| Stainless Steel Beads, 5 mm | Qiagen | 69989 |
| SuperFrost microscope slides | epredia | AA00008132E01MNZ10 |
| TissueLyser | Qiagen, Hilden, Germany | |

## Viruses, plaque, and neutralization assays

The LCMV strain Docile was originally obtained from Dr. C. J. Pfau (Troy, NY). LCMV Armstrong was kindly provided by Rolf Zinkernagel, University of Zurich, Zurich, Switzerland. Both strains were propagated in L929 cells (Xu et al, 2021). Viral titers were assessed using a plaque-forming assay as previously described (Lang et al, 2013). Briefly, tissue was homogenized in Hank's balanced salt solution using a TissueLyser (Qiagen, Hilden, Germany). Serial dilutions of samples were added to $8 \times 10^5$ MC57 fibroblasts in 24-well plates and incubated at 37 °C for 3 h after which the cells were coated with 1% methylcellulose containing medium. Forty-eight hours later, the plates were fixed with 4% formalin, permeabilized (1% Triton X-100 in HBSS), and stained with the anti-LCMV NP antibody (clone VL-4, produced in-house) and an ECL-conjugated anti-rabbit-IgG secondary antibody. For the detection of LCMV-specific neutralizing antibodies (nABs), serial dilutions of plasma samples were added to 96-well culture plates and any remaining viral load was inactivated by UV light. In all, $3 \times 10^2$ PFU LCMV Docile was added to each well and incubated for 90 min at 37 °C. LCMV detection was performed as described for the plaque assay. VSV (Indiana strain, Mudd-Summers isolate) was originally obtained from Professor D. Kolakofsky (University of Geneva, Switzerland) and was propagated on BHK-21 cells (Khairnar et al, 2015).

## Mice

*Pax5*$^{-/+}$ and *Pax5*$^{+/+}$ (WT) backcrossed to C57BL/6J mice for at least 10 generations were maintained under specific pathogen-free

(SPF) conditions. For all experiments involving *Pax5*$^{-/+}$ heterozygous mice, *Pax5*$^{+/+}$ (WT) littermate controls were used. For infection experiments, mice were transferred to conventional facility (CF) and intravenously injected with $10^6$ PFU of LCMV Docile. For immune training experiments, 1 mg of β-glucan (InvivoGen, tlrl-bgp) was injected seven prior to infection with $10^6$ PFU of LCMV Docile. For immune training experiments without *Pax5* heterozygotes, C57BL/6J mice were used (purchased from Charles River). Experiments were performed under the authorization of LANUV in accordance with German laws for animal protection.

## Histology and CO-detection by indEXing (CODEX)

Histological analysis of snap-frozen or paraffin-embedded tissue sections using hematoxylin and eosin (H&E) staining was previously described (Xu et al, 2022). Images were acquired with the Aperio slide scanner (Leica, Nussloch, Germany) and digitalized. Quantification of infiltrates was done using Qupath-0.5.0 software. Specifically, liver slides were trained with pixel classifier Random trees (RTrees), and the whole liver was divided by background, healthy, and infiltrated sections. The ratio of infiltration area/healthy area was evaluated and graphed by Prism GraphPad. CODEX analysis on splenic fresh-frozen (FF) tissue was adapted from previously described applications (Schniederjohann et al, 2025). Briefly, FF 5-μm sections were placed on SuperFrost® Plus objective slides (Thermo Scientific) and kept at −20 °C until staining. Tissue retrieval was performed with Dry-rite beads (Thermo Fisher Scientific) and immersed in acetone before hydration in Akoya hydration buffer and fixation with 16% paraformaldehyde (PFA). Tissues were equilibrated in staining buffer (Akoya, CODEX staining kit) for 30 min. The antibody cocktail was prepared in blocking buffer. Slides were stained with antibody cocktail overnight at 4 °C in a humidity chamber. Post staining antibody fixation was performed according to Akoya User Guide. Tissues were bleached after fixation using an LED panel (4100-4600LUX, AGPTEK). Reporter solution was added to the reporter plate, and multicycling was performed with the PhenoCycler (Akoya Biosciences). The slides were imaged using an immunofluorescent microscope (Phenocycler Fusion (Akoya Biosciences)). CODEX staining cycle info, antibody dilutions and oligonucleotide sequences are provided in Table EV1. The tissues were afterwards H&E stained and also imaged. Image processing was performed using the CODEX processor and QuPath-0.5.1 software.

## Serum biochemistry

Lactate dehydrogenase (LDH) was measured using the automated biochemical analyzer SPOTCHEM EZ SP-4430 (ARKRAY, Amstelveen, the Netherlands) using the Spotchem II Liver-1 strips.

## Flow cytometry

For surface staining, singly suspended cells were incubated with antibodies (anti-CD19, CD3, CD4, CD8, CD138, CD28, B220, IL-7r, PD1, MHC-II, CD21, CD23, CD25 (1:50), PD-L1, CD11b, IgM, IgD, Ly6C, Ly6G, Sca-1, CD117, CD150, CD48 or the Hematopoietic Lineage Antibody Cocktail). Antibodies were diluted 1:300 in FACS buffer for all staining panels performed unless indicated otherwise.

DAPI was added (1:10,000) to the samples prior to measurements. Tetramer and intracellular cytokine staining were performed as previously described (Xu et al, 2022). Briefly, singly suspended cells were incubated with tetramer-gp33 or tetramer-np396 (1:100) for 15 min at 37 °C followed by incubation with surface antibodies (anti-CD8, CD3, PD1, IL-7r, Tim-3, CD4, LAG3, and CD62L) for 30 min at 4 °C. For intracellular cytokine restimulation (ICS) staining, singly suspended cells were stimulated with the gp33 LCMV-specific peptide for 1 h followed by the addition of Brefeldin A (Thermo Fisher Scientific) for another 5 h at 37 °C. This was followed by subsequent staining with anti-CD8, anti-CD4, anti-KLRG1, anti-CD25 and fixable viability dye (1:100). After surface staining, cells were fixed and permeabilized for intracellular staining using the Foxp3/ Transcription Factor Staining Buffer Set (Thermo Fisher Scientific) and stained with anti-interferon γ (IFN-γ), anti-Granzyme B and anti-FOXP3 antibodies. Flow cytometry analysis for peripheral lymphocytes was performed as previously described (Escudero et al, 2022). Experiments were acquired using Cytoflex or Fortessa and analyzed using FlowJo software.

## LCMV-glycoprotein GP-1-specific IgG measurements

GP-1-specific IgG using ELISA was done as previously described (Khairnar et al, 2015). Briefly, after coating 96-well flat-bottom plates with anti-human IgG (Jackson ImmunoResearch Laboratories, Inc.) for 24 h at 4 °C, plates were washed and blocked with 2% FCS in PBS for 2 h followed by incubation for 3 h with LCMV Gp-Fc supernatant (produced in-house) at room temperature. After washing, plates were titrated and incubated with 1:30 pre-diluted serum over four wells with 1:2 dilutions for 90 min and subsequently incubated with HRP-conjugated anti-mouse–IgG antibody (Sigma). Titers are presented as threefold dilution steps (log3) times the predilution (×30).

## Immunoglobulin quantification in mouse plasma and multiplex cytokine analyses

Immunoglobulin levels were quantified in mouse heparin-anticoagulated plasma samples using the Mouse Immunoglobulin Isotyping Panel(6-plex) with V-bottom Plate kit (BioLegend) according to the manufacturer's instructions. Briefly, plasma samples were diluted 100,000–200,000-fold based on preliminary experiments to determine the optimal dilution range. The diluted samples were incubated with the LEGENDPlex™ capture beads, detection antibodies, and standards. After washing, the samples were analyzed on an Attune NxT Flow Cytometer (Thermo Fisher Scientific). Data was analyzed using the LEGENDPlex™ software (BioLegend). The levels of GM-CSF, GROα, IFN-α, IFN-ß, IFN-γ, IL-1ß, IL-10, IL-12p70, IL-18, IL-4, IL-6, IP-10, MCP-1, MIG, MIP-1α, MIP-1β, RANTES, TNF-α and VEGF-A were assessed using the Mouse ProcartaPlex Mix&Match 19-Plex (Invitrogen) using the Bio-Plex200 (Bio-Rad Laboratories, Hercules, CA, USA) according to the manufacturer's instructions.

## Graphics

A synopsis image was created with BioRender.com as where other graphical images in the figures.

## The paper explained

### Problem

While viral infections can incur a serious health burden, their effects have not been well-elucidated in hosts genetically predisposed to B-ALL development particularly those with mild immunodeficiencies.

### Results

In this study, we used a Pax5 heterozygous ($Pax5^{-/+}$) mouse model mimicking human *PAX5* germline alterations to show that these mice were more susceptible to the effects of a chronic LCMV infection. Although early CD8$^+$ T-cell responses were robust in *Pax5* haploinsufficient hosts following chronic LCMV infection, $Pax5^{-/+}$ mice were deficient in the ability to produce LCMV-specific neutralizing antibodies compared to their control hosts. This led to impaired long-term viral clearance and increased elevated lactate dehydrogenase levels. We then showed that susceptibility to chronic infection in the mildly immunocompromised host can be attenuated by prophylactic pre-treatment with the immune trainer β-glucan. β-glucan pre-treatment improved viral clearance, CD8$^+$ T-cell immunity, neutralizing antibody production, and decreased monocyte immunosuppression in LCMV-resident organs.

### Impact

This work highlights the ability of β-glucan to reverse the deleterious effects of a chronic viral infection in susceptible hosts in the prophylactic and albeit less effectively, in the interventional setting.

## Statistical analyses

Analyses were not blinded. Mice were randomly divided into different treatment groups. Statistical analysis was GraphPad software and a *P* value of less than 0.05 was considered statistically significant. Exact *P* values that were significant are indicated in the figures or figure legends. A two-tailed unpaired Student's *t* test was used to compare two conditions and a one-way ANOVA with a Tukey or Dunnett post hoc test was used to compare multiple groups. Statistical analysis of survival curves was performed using a log-rank (Mantel–Cox) test with a Bonferroni correction for multiple curve comparisons. A Fisher's exact test was used to analyze contingencies. Data are represented by the mean and standard error of the mean (mean ± SEM).

## Data availability

This study includes no data deposited in external repositories.

The source data of this paper are collected in the following database record: biostudies:S-SCDT-10_1038-S44321-025-00208-4.

## Peer review information

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

## Acknowledgements

AAP, UF, and AB acknowledge support from the Deutsche José Carreras Leukämie-Stiftung (DJCLS 18R/2021, DJCLS 07/19). AAP and UF are supported by the German Childhood Cancer Foundation (A2023/31). AAP, UF, and AB acknowledge support from the German Federal Office for Radiation Protection (BfS). AAP is supported by the DZIF (TTU 07-711). AAP, UF, SJ, and AB acknowledge support from the German Ministry for Education and Research (Bundesministerium für Bildung und Forschung BMBF)— grant no. 01KD2410A (EDI-4-ALL). AB and UF acknowledge support from the Katharina-Hardt-Stiftung, the Christiane und Claudia Hempel Stiftung, and Löwenstern e.V. (FKZ: 3618S32275 and 3618S32274). UF is supported by the Deutsche Forschungsgemeinschaft (DFG, German Research Foundation—Projektnummer 495318549) and the German Cancer Aid (Priority Program Cancer Prevention —Graduate School). JH acknowledges support from the ERC Stg 85222 PreventALL and German Cancer Aid (DKH 70114539). ZL is funded by the Chinese Scholarship Council (CSC). SK and OS are funded by the Düsseldorf School of Oncology (DSO). FA acknowledges support from the Deutsche Forschungsgemeinschaft (DFG, German Research Foundation, 547295412). SD acknowledges support from the BMBF through the ERA-NET TRANSCAN grant BIALYMP (01KT2311), the BMBF ERA PerMed grant SYMMETRY (01KU2210), the Hairy Cell Leukemia Foundation and the Else Kröner Fresenius Foundation. P-MB acknowledges support from the Else Kröner Fresenius Foundation and the Young Investigator Grant of the Medical Faculty Düsseldorf. GH acknowledges support from the Deutsche Forschungsgemeinschaft (DFG, German Research Foundation) under Germany's Excellence Strategy (EXC2151–390873048), DFG TRR237, DFG TRR259, and by the German Center for Infectious Diseases (DZIF) (TTU 07.834). PAL acknowledges support from the German Research Council (DFG, LA2558/8-1, RTG1949), the Jurgen Manchot Foundation (Molecules of Infection), the Volkswagen Foundation, and the Medical Faculty of the Heinrich-Heine University (Forschungskommission, Dusseldorf School of Oncology).

## Author contributions

**Zhe Lu**: Conceptualization; Data curation; Formal analysis; Validation; Investigation; Methodology; Writing—review and editing. **Olivia Stencel**: Conceptualization; Data curation; Formal analysis; Validation; Investigation; Methodology; Writing—review and editing. **Wei Liu**: Conceptualization; Data curation; Formal analysis; Validation; Investigation; Methodology; Project administration; Writing—review and editing. **Eleni Vasileiou**: Data curation;

Formal analysis; Investigation; Methodology. **Haifeng C Xu**: Resources; Methodology. **Piyush Pandey**: Investigation; Methodology. **Paweł Stachura**: Investigation; Methodology. **Abdelrahman Elwy**: Investigation; Methodology. **Anastassia Tsombal**: Investigation; Methodology; Project administration. **Ann-Sophie Mai**: Investigation; Methodology. **Franziska Auer**: Conceptualization; Resources. **Mina N F Morcos**: Conceptualization; Resources. **Maximilian Seidl**: Data curation; Software; Methodology. **Sarah Koziel**: Investigation; Methodology. **Peter-Martin Bruch**: Resources; Software; Investigation; Methodology. **Sascha Dietrich**: Resources; Software; Investigation; Methodology. **Sarah Elitzur**: Resources. **Gunther Hartmann**: Resources. **Karl S Lang**: Resources; Methodology. **Stefan Janssen**: Investigation. **Ute Fischer**: Resources; Project administration. **Sanil Bhatia**: Resources; Project administration. **Philipp A Lang**: Resources. **Arndt Borkhardt**: Conceptualization; Resources; Supervision; Funding acquisition; Investigation; Writing—original draft; Project administration; Writing—review and editing. **Julia Hauer**: Conceptualization; Resources. **Aleksandra A Pandyra**: Conceptualization; Resources; Data curation; Software; Formal analysis; Supervision; Funding acquisition; Validation; Investigation; Visualization; Methodology; Writing—original draft; Project administration; Writing—review and editing.

Source data underlying figure panels in this paper may have individual authorship assigned. Where available, figure panel/source data authorship is listed in the following database record: biostudies:S-SCDT-10_1038-S44321-025-00208-4.

## Funding

## Disclosure and competing interests statement

PAL, KSL, HCX, and PP declare that they are involved in the development of LCMV for clinical application in oncology in cooperation with, as founders of, or as advisors to Abalos Therapeutics GmbH. The remaining authors declare no competing interests.

# Expanded View Figures

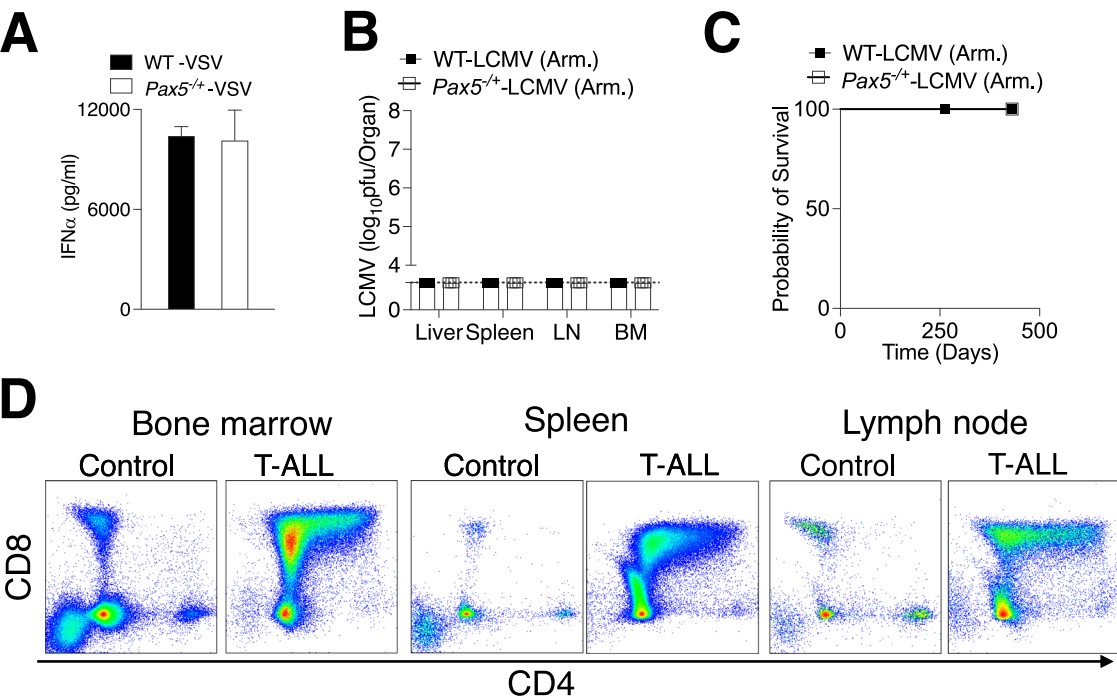

**Figure EV1. Pax5 haploinsufficiency does not compromise hosts following VSV and acute LCMV infection.**

(A) WT and $Pax5^{-/+}$ mice were infected with $10^6$ PFU of VSV and IFN-α concentration was determined in the plasma of WT ($n = 3$ mice per group) and $Pax5^{+/-}$ mice ($n = 4$ mice per group) 24 h after infection. WT and $Pax5^{-/+}$ mice were infected with $10^6$ PFU of LCMV-Armstrong strain (Arm.) and (B) viral titers were determined 15 days post infection using the plaque assay ($n = 3$ mice per group). Statistical analyses were performed using a Student´s $t$ test (unpaired, two-tailed). (C) Survival was monitored ($n = 3$ mice per group). Statistical analysis was performed a Log-rank (Mantel–Cox) test with a Bonferroni correction for comparisons. (D) FACS blots of spleen, bone marrow and lymph node tissue showing the T-ALL that developed at day 182 post LCMV infection in a $Pax5^{-/+}$ mouse. Error bars indicate SEM. Source data are available online for this figure.

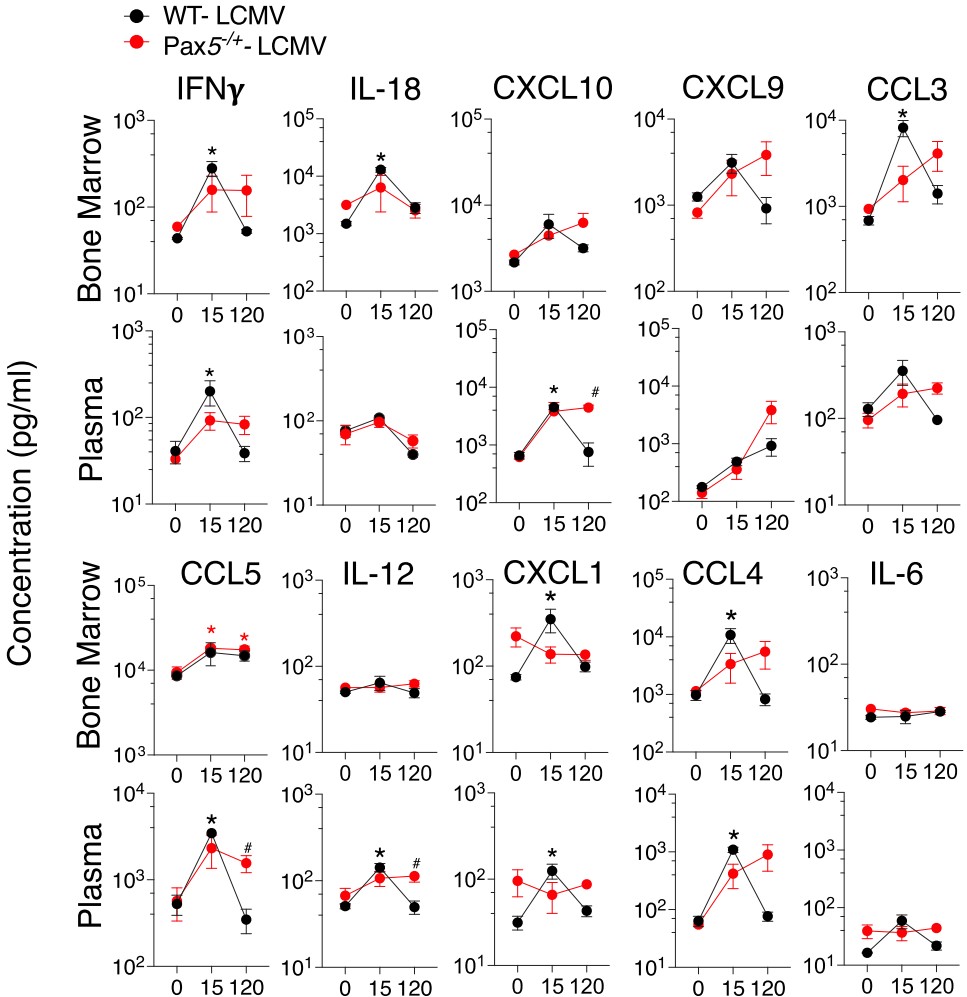

**Figure EV2. Pax5 haploinsufficiency shapes distinct cytokine profiles in the plasma and bone marrow microenvironment in response to chronic infection.**

Cytokine levels in the bone marrow and plasma were evaluated using the Procarta 19-Plex assay in uninfected naïve controls (day 0), day 15 and day 120 post infection with $10^6$ PFU of LCMV Docile in *Pax5$^{-/+}$* and WT mice ($n = 3$ mice per group). Concentrations in pg/ml of detectible cytokines are shown. *As indicated in the figure represents significant differences relative to time point 0 within each genotype as determined by a one-way ANOVA with Dunnett's post hoc test. IFNγ (BM $P = 0.0036$, plasma $P = 0.045$), IL-18 ($P < 0.0001$), CXCL10 ($P = 0.0002$), CCL3 ($P = 0.004$), CCL5 (BM day 15 $P = 0.0105$, day 120 $P = 0.0155$, plasma $P < 0.0001$), IL-12 ($P = 0.0018$), CXCL1 (BM $P = 0.0358$, plasma $P = 0.0081$), CCL4 (BM $P = 0.0150$, plasma $P < 0.0001$). #As indicated in the figure represents significant differences between the *Pax5$^{-/+}$* and WT group at a given time point as determined by a Student's *t* test (unpaired, two-tailed). CXCL10 ($P = 0.003$), CCL5 ($P = 0.030$), IL-12 ($P = 0.028$). Error bars indicate SEM. Source data are available online for this figure.

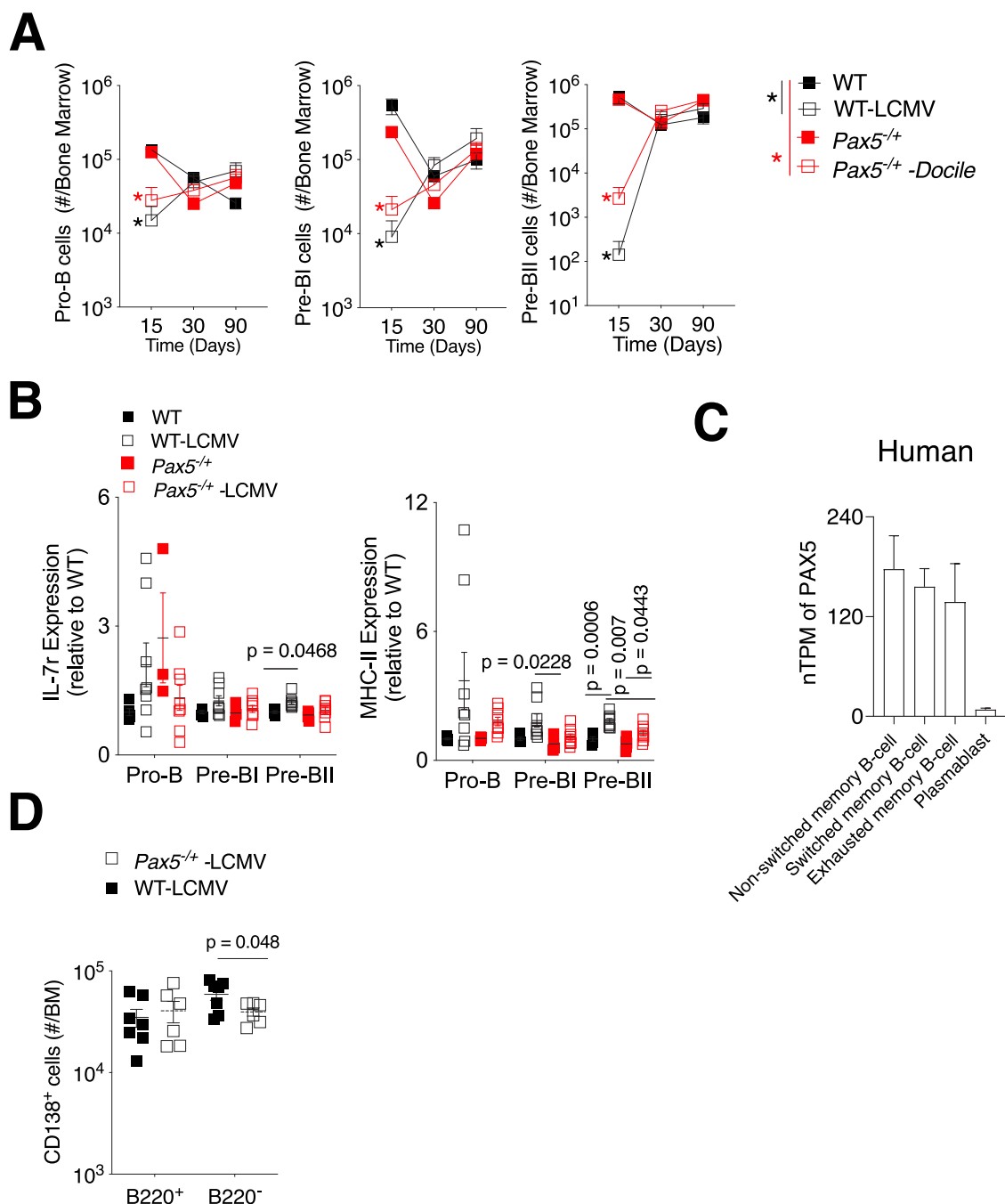

**Figure EV3. Chronic LCMV infection affects early B cells populations in the bone marrow of *Pax5^-/+* and WT hosts.**

*Pax5^-/+* and WT mice were intravenously infected with 10^6 PFU of LCMV Docile. (A) Pro-B, Pre-BI and Pre-BII cells were measured in the bone marrow at the indicated days post infection using FACS ($n \geq 3$ mice per group). (Pro-B, WT $P = 0.0007$ *Pax5^-/+* $P = 0.0010$), (Pre-BI, WT $P = 0.001$ *Pax5^-/+* $P = 0.0007$), (Pre-BII, WT $P = 0.0055$ *Pax5^-/+* $P = 0.0037$). (B) Surface molecule MHC-II and IL-7r expression was measured on Pro-B, Pre-BI and Pre-BII cells in the bone marrow 90 days post infection ($n \geq 3$ mice per group). Statistical analyses for (A, B) were performed using a one-way ANOVA with a Tukey post hoc test. (C) PAX5 expression data in different human B-cell subsets was mined from The Human Protein Atlas, Monaco dataset ($n = 4$). (D) B220^+CD138^+ and B220^-CD138^+ numbers were evaluated in the bone marrow of WT ($n = 7$ mice per group) and *Pax5^-/+* mice ($n = 6$) 120 days post infection using FACS analysis. Error bars indicate SEM. Statistical analysis was carried out using a Student´s t test (unpaired, two-tailed). Source data are available online for this figure.

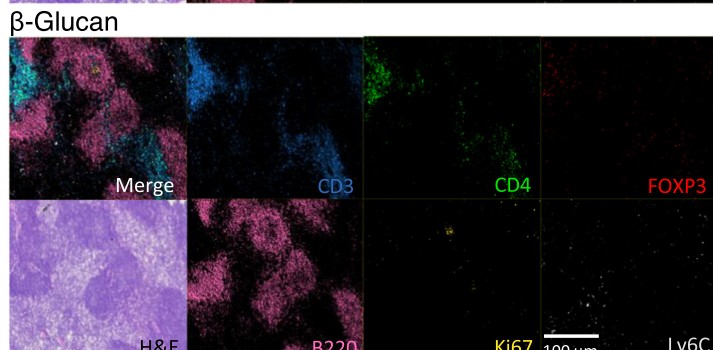

A

B

C

D

Control

β-Glucan

◀ **Figure EV4. Immune training with β-glucan improves T-cell immunity following chronic LCMV infection.**

(A) WT mice were injected with 1 mg of β-glucan and LSK (Lin⁻Sca-1⁺c-KIT⁺) cells were measured in the bone marrow 24 h and 15 days post treatment ($n \geq 3$ mice per group). (B) WT mice were injected with 1 mg of β-glucan and 7 days later this was followed by infection with $10^6$ PFU of LCMV Docile. LCMV virus titers were determined in the kidney, lymph node (LN) and liver at day 15 ($n = 3$ mice per group) and 30 ($n \geq 7$ mice per group) days post LCMV infection using plaque assay. Statistical analysis of the contingency table was done using a Fisher's exact test. (C) Numbers of B1, marginal zone (MZ), follicular (FO) and transitional (T1, T2 and T3) B cells were assessed in naive ($n = 4$ mice per group) and β-glucan ($n = 5$ mice per group) treated mice at day 30 post β-glucan treatment. (D) Splenic sections were stained with the indicated antibodies and detected using CO-Detection by indEXing (CODEX), scale bar indicates 100 μm; (a representative image of $n$ of 5 (mice per group) is shown). Error bars indicate SEM. Unless otherwise stated, statistical analyses were performed using a Student's t test (unpaired, two-tailed). Source data are available online for this figure.

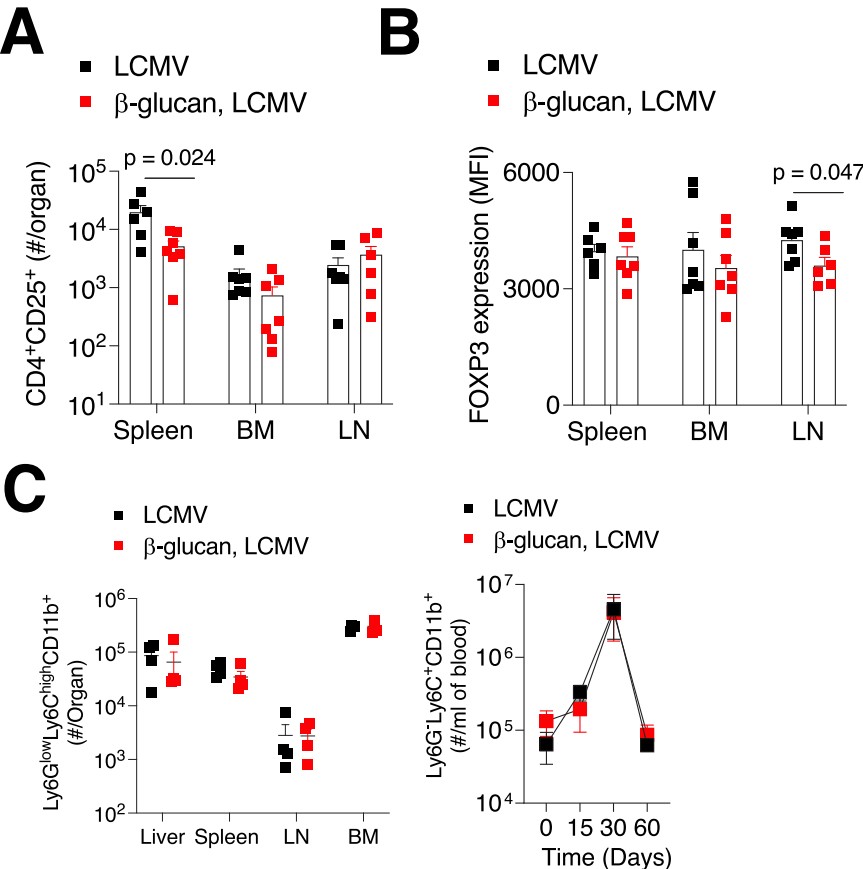

**Figure EV5. Immune training with β-glucan affects immunosuppressive infiltrates following chronic LCMV infection.**

WT mice were injected with 1 mg of β-glucan and 7 days later this was followed by infection with $10^6$ PFU of LCMV Docile. (A) 30 days post infection, numbers of Treg's (CD4$^+$CD25$^+$) as well as (B) Treg FOXP3 expression were measured using FACS analysis in the spleen, bone marrow (BM) and lymph node (LN) of infected mice ($n \geq 6$ mice per group). (C) 30 Days post infection, frequencies of monocytes (CD11B$^+$Ly6G$^{low}$Ly6C$^{high}$) were measured using FACS analysis in the spleen, bone marrow (BM) and lymph node (LN), liver ($n = 4$ mice per group) and in the blood at the indicated time points ($n \geq 3$ mice per group). Error bars indicate SEM. Error bars indicate SEM. Statistical analyses were performed using a Student´s $t$ test (unpaired, two-tailed). Source data are available online for this figure.

