## [Peer Review File · EMBO Molecular Medicine]

Immune training enhances anti-viral responses in Pax5-/+ mice susceptible to chronic infection

Zhe Lu, Olivia Stencel, Wei Liu, Eleni Vasileiou, Haifeng Xu, Piyush Pandey, Pawel Stachura, Abdelrahman Elwy, Anastassia Tsombal, Ann-Sophie Mai, Franziska Auer, Mina Morcos, Maximilian Seidl, Sarah Koziel, Peter-Martin Bruch, Sascha Dietrich, Sarah Elitzur, Gunther Hartmann, Karl Lang, Stefan Janssen, Ute Fischer, Sanil Bhatia, Philipp Lang, Arndt Borkhardt, Julia Hauer, and Aleksandra Pandyra

Corresponding author: Aleksandra Pandyra (aleksandra.pandyra@uni-duesseldorf.de)

Review Timeline:

Submission Date:	31st May 24
Editorial Decision:	4th Jul 24
Revision Received:	26th Dec 24
Editorial Decision:	24th Jan 25
Revision Received:	19th Feb 25
Accepted:	20th Feb 25

Editor: Zeljko Durdevic

Transaction Report:

4th Jul 2024

Dear Prof. Pandya,

Thank you for the submission of your manuscript to EMBO Molecular Medicine. We have now received feedback from the three reviewers who agreed to evaluate your manuscript. All three referees recognize interest of the study but also raise important and partially overlapping concerns that should be addressed in a major revision. If you would like to discuss further the points raised by the referees, I am available to do so via email or video. Let me know if you are interested in this option.

We would welcome the submission of a revised version within three months for further consideration. Please let us know if you require longer to complete the revision.

I look forward to receiving your revised manuscript.

Yours sincerely,

Zeljko Durdevic

We require:

- 1) A .docx formatted version of the manuscript text (including legends for main figures, EV figures and tables). Please make sure that the changes are highlighted to be clearly visible.
- 2) Individual production quality figure files as .eps, .tif, .jpg (one file per figure). For guidance, download the 'Figure Guide PDF': (<https://www.embopress.org/page/journal/17574684/authorguide#figureformat>).
- 3) A .docx formatted letter INCLUDING the reviewers' reports and your detailed point-by-point responses to their comments. As part of the EMBO Press transparent editorial process, the point-by-point response is part of the Review Process File (RPF), which will be published alongside your paper.
- 4) A complete author checklist, which you can download from our author guidelines (<https://www.embopress.org/page/journal/17574684/authorguide#submissionofrevisions>). Please insert information in the checklist that is also reflected in the manuscript. The completed author checklist will also be part of the RPF.
- 5) Please note that all corresponding authors are required to supply an ORCID ID for their name upon submission of a revised manuscript.
- 6) It is mandatory to include a 'Data Availability' section after the Materials and Methods. Before submitting your revision, primary datasets produced in this study need to be deposited in an appropriate public database, and the accession numbers and

database listed under 'Data Availability'. Please remember to provide a reviewer password if the datasets are not yet public (see <https://www.embopress.org/page/journal/17574684/authorguide#dataavailability>).

13) Author contributions: You will be asked to provide CRediT (Contributor Role Taxonomy) terms in the submission system. These replace a narrative author contribution section in the manuscript.

14) A Conflict of Interest statement should be provided in the main text.

15) Every published paper now includes a 'Synopsis' to further enhance discoverability. Synopses are displayed on the journal

webpage and are freely accessible to all readers. They include a short stand first (maximum of 300 characters, including space) as well as 2-5 one-sentences bullet points that summarizes the paper. Please write the bullet points to summarize the key NEW findings. They should be designed to be complementary to the abstract - i.e. not repeat the same text. We encourage inclusion of key acronyms and quantitative information (maximum of 30 words / bullet point). Please use the passive voice. Please attach these in a separate file or send them by email, we will incorporate them accordingly.

16) Include a Reagents and Tools Table as part of the Methods section, which can be downloaded from our author guidelines (<https://www.embopress.org/page/journal/17574684/authorguide#structuredmethods>)

**** Reviewer's comments ****

Referee #1 (Remarks for Author):

The manuscript entitled, "Immune training enhances anti-viral responses in Pax5-/+ mice susceptible to chronic infection" reported the critical role of PAX-5 in controlling chronic virus LCMV infection. The authors found that WT and PAX5-/+ mice showed no difference with VSV infection. However, Pax5-/+ mice were more susceptible to LCMV infection. This phenotype may be related to decreased neutralizing antibody production and increased exhausted CD8 T cells. Additionally, the authors used well known trained immunity inducer beta-glucan to show that induction of trained immunity by beta-glucan enhances viral clearance, CTL immunity and reduced monocyte immunosuppression. Although these data have translational potential, there are some concerns in this study.

1. The authors claimed that induction of trained immunity leads to chronic virus infection control. However, these studies mainly looked at T cells and monocytes. Did beta-glucan treatment impact on neutralizing antibody production and B cell compartment in the bone marrow?
2. The study lacks in-depth mechanistic studies. For example, it is unknown how trained immunity impacts on CTL activity and function.
3. Figure 1. It is unclear which tissue was analyzed for panel D. For panel E, only liver showed viral titer difference.
4. Figure 2. Panel E needs to be quantitated.
5. Figure 4. The authors only used PD-1 marker and claimed that these T cells are exhausted. More markers such as TIM3 and LAG3 need to be added.
6. Figure 5. Statistical analysis is needed for panel C.
7. Figure 6. Panel A. It is unclear whether there is any statistical significance for these data although in the text the authors claimed significant difference.

Referee #2 (Comments on Novelty/Model System for Author):

The manuscript used appropriate methodology and there are no issues ethics or model organisms

Referee #2 (Remarks for Author):

This is an interesting manuscript showing that β -glucan administration affects the outcome of LCMV infection in PAX5-/+ mice. Major comments

1. Did the authors observe any leukemia development in b-glucan injected mice. B-glucan induces inflammation and this could stimuli leukemogenesis.
2. Administration of adjuvants is known to induce antibody production. B-glucan acts as an adjuvant driving innate immune responses. Did the authors observe any difference in antibody production upon b-glucan administration?
3. Did the authors inject b-glucan after initiation of LCMV infection. This could be more relevant for clinical practice.
4. There is a difference in plasma cells in Pax5-/+ mice. This is due to an effect on survival of plasma cells or differentiation of plasmablasts to plasma cells? Is Pax5 expressed in these cells?

Minor comment

1. Figure 1 "of LCMV Docile and (B) survival was monitored" it should be (C)

Referee #3 (Comments on Novelty/Model System for Author):

The model is necessarily a mouse model of PAX5^{+/-} leukemia and although quite preliminary may be extended to childhood BCP-ALL and perhaps augment its prevention

Referee #3 (Remarks for Author):

The article from Lu et al uses a PAX5^{+/-} mouse model to study the responses to certain viral stimuli and the subsequent changes in survival outcome observed after treating animals with a chronic infection. After LCMV infection, these animals do succumb to infection and also reveal a distinct BM environment and cytokine picture.

The authors assess the ability of beta-glucan via an immune training approach to reverse the effects of PAX5^{+/-} animals susceptible to LCMV infection by increasing viral clearance and showing improved T-cell immunity. Interestingly, these animals now do show decreased susceptibility to infection.

The manuscript sits within the Aims and Scope of the journal and is sound and well written.

1. It is unclear from the manuscript exactly what these mice are succumbing to. There is no mention of leukemia in the results? Do the animals have pB-ALL? What is the phenotype of the B cells and are they clonal? Indeed, why did one animal develop T-ALL? Was that expansion shown to be clonal?

2. What is the latency period for PAX5^{+/-} susceptibility to chronic infection as opposed to moving these animals to a conventional facility? Is this period now shorter?

3. What is the current health monitoring report for this SPF facility - could other viruses already be present?

4. Given that the effect of VSV was not enough to compromise these animals, why didn't the authors use an acute strain of LCMV as a direct comparison? Is this response somehow peculiar to VSV?

Minor

5. Was the induced cytokine pattern the same for all animals that did or did not succumb to chronic infection? Regardless of B or T cell ALL?

6. What was the effect on LSK cell numbers after injecting beta-glucan alone into PAX5^{+/-} animals?

We would like to thank the reviewers for their extremely helpful comments and the time they took to review the manuscript. Their excellent suggestions have significantly strengthened the manuscript. Please see the detailed responses below:

Referee #1 (Remarks for Author):

The manuscript entitled, "Immune training enhances anti-viral responses in Pax5-/+ mice susceptible to chronic infection" reported the critical role of PAX-5 in controlling chronic virus LCMV infection. The authors found that WT and PAX5-/+ mice showed no difference with VSV infection. However, Pax5-/+ mice were more susceptible to LCMV infection. This phenotype may be related to decreased neutralizing antibody production and increased exhausted CD8 T cells. Additionally, the authors used well known trained immunity inducer beta-glucan to show that induction of trained immunity by beta-glucan enhances viral clearance, CTL immunity and reduced monocyte immunosuppression. Although these data have translational potential, there are some concerns in this study.

1. The authors claimed that induction of trained immunity leads to chronic virus infection control. However, these studies mainly looked at T cells and monocytes. Did beta-glucan treatment impact on neutralizing antibody production and B cell compartment in the bone marrow?

We agree with the reviewer that we should also show effects of β -glucan on the B-cell compartment in the bone marrow and on neutralizing antibody production. In the context of pre-B cells, the most prominent effects occurred at Day 15 post LCMV-infection (old Supplementary Figure 3B, new Supplementary Figure 3A) where pre-B subsets (pro-B, pre-BI and pre-BII) were significantly depleted in the infected BM of both Pax5^{+/+} and WT mice. As the numbers had stabilized by day 30 relative to uninfected controls in both genotypes, we assessed pre-B cell subsets at day 15 post-LCMV infection in the presence of β -glucan. β -Glucan was partially able to restore the numbers of pre-B cell subsets to alleviate the infection-driven depletion. This new data is now presented in new Figure 6C. Furthermore, as suggested by the reviewer, we now show the effects of β -glucan on neutralizing antibody production at day 30 and 60 post-infection (New Figure 6B). Specifically, β -glucan significantly increased LCMV-neutralizing antibody production at Day 30 post-infection. All new results are incorporated and discussed in main body of the text marked in red.

2. The study lacks in-depth mechanistic studies. For example, it is unknown how trained immunity impacts on CTL activity and function.

In response to the reviewer's comment, we have carried out additional experiments to address this issue. In New Figure 7F-J we infected mice with a lower dose of LCMV-Docile (10^4 PFU in order to enable ex vivo restimulation with LCMV-specific gp33 peptide at day 30 post-infection) in combination with pre-treatment of β -glucan (Figure 5B) and post-infection treatment (New Figure 7A). We now show that at day 30 post-infection, there was significantly increased granzyme B expression in CD8⁺ T cells upon re-stimulation with the LCMV-specific gp33 peptide

in the β -glucan pre-treated infected group relative to the post-treated β -glucan infected and infected alone group (New Figure 7G) in the lymph node and liver. This was particularly relevant in the lymph node where the most consistent reduction in monocyte PD-L1 expression occurred in the β -glucan pre-treated group (New Figure 7F) suggesting a direct de-repression of CTL function through the PD1/PD-L1 axis. The co-localization areas of CD3 and Ly6C, as well as Ly6C and B220 in lymphoid tissue (New Figure 7J) support a probable interaction not only between monocytes and T cells but monocytes and B cells. KLRG1 expression was also increased in CD8⁺ T cells in the spleen and liver of the β -glucan pre-treated group (New Figure 7H) both organs where there was a higher percentage of tet⁺ CTL cells (New Figure 7I). We have also generated additional data to indicate how β -glucan pre-treatment could alleviate chronic inflammatory signals that could exhaust both monocytes and CTLs by demonstrating decreased levels of pro-inflammatory chemokines (CCL3 and CCL5) and cytokines (CXCL9 and CXCL10) at day 15 post chronic infection in the β -glucan pre-treated group (New Figure 6H). Indeed, at day 15, at the higher 10⁶ PFU LCMV Docile doses only the splenic CD8⁺T cells harvested from the β -glucan pre-treatment were capable of being re-stimulated with the gp33 peptide to increase interferon gamma production (Now Figure 6E). We acknowledge that improved CTL immunity as a consequence of β -glucan pre-treatment is a complex phenomenon supported by additional effects on other immune cells including B cells leading to for example improvements in neutralizing antibody production (New Figure 6B). As β -glucan treatment alone did not lead to increases in B cell numbers in the bone marrow (New Figure 6C) or spleen (New Supplementary Figure 4C) this improvement could be due to increased B cell fitness and decreased monocyte immunosuppression during infectious perturbation.

3. Figure 1. It is unclear which tissue was analyzed for panel D. For panel E, only liver showed viral titer difference.

We thank the reviewer for this important point and we agree that this is not clear enough. Firstly, to increase the clarity, in Figure 1D the identity of tissue analyzed is clearly indicated below the graph. Secondly, to simplify the data presented in previous Figure 1E (Now new Figure 2C) and to emphasize our point that Pax5^{+/+} mice experience slower viral clearance we have carried out additional experiments to increase our numbers of mice at day 120 post-infection. We now show that while at day 15 post infection there were no differences in LCMV viral titres between Pax5^{+/+} and WT mice, Pax5^{+/+} mice were characterized by significantly higher virus titres in the spleen and lymph node 120 days post-infection. By day 120 post infection, many of the WT mice had cleared the virus. This data is now presented in new Figure 2C. This is in line with the subsequent data showing increased serum LDH levels and increased infiltrates in the liver of the Pax5^{+/+} mice at day 120 post-infection in Figure 2D and 2E respectively.

4. Figure 2. Panel E needs to be quantitated.

Using deep learning algorithm from the Qupath-0.5.0 software, we now quantify the infiltration in liver tissue of infected and uninfected Pax5^{+/+} and WT mice. The quantification is shown in Figure 2E beside representative images and is now

described in Materials and Methods. Pax5^{-/+} mice had significantly increased levels of infiltrates at day 120 post infection compared to the WT injected controls.

5. Figure 4. The authors only used PD-1 marker and claimed that these T cells are exhausted. More markers such as TIM3 and LAG3 need to be added.

As suggested by the reviewer, we now show LAG3 and TIM-3 exhaustion markers (New Figure 4C) on tetramer positive cells in the bone marrow of LCMV infected Pax5^{-/+} mice compared to infected WT controls.

6. Figure 5. Statistical analysis is needed for panel C.

We thank the reviewer for the comment; as pointed out, we corrected this issue and included the statistical significance in Figure 5C. As stated in the text and now in the figure there were no significant differences between the indicated groups. Even without the Bonferroni correction for multiple curve comparisons, when comparing two curves at a time, the P- value when we compare Pax5^{-/+}-β-Glucan-Docile vs WT-β-Glucan- Docile is 0.066. The P- value when we compare Pax5^{-/+}-β-Glucan- Docile vs Pax5^{-/+}-β-Glucan is 0.1320.

7. Figure 6. Panel A. It is unclear whether there is any statistical significance for these data although in the text the authors claimed significant difference.

Again, this is an excellent point. Indeed, in the previous old Figure 6A, when viral titres (as assessed using plaque assay) were compared, there was no statistical significance between the β-glucan-LCMV and LCMV group. However, a higher proportion of the mice in the β-glucan pre-treated group had cleared the virus by day 30 and we represented this faster clearance using a contingency graph (old Figure 6B) which is where the claim of significance stemmed from. However, we agree that this is confusing. We have now added more n's and show the significant viral load results where there was a significant decrease in the spleen of β-glucan pre-treated mice at day 30 post infection (New Figure 6A). The rest of the organs we have moved to supplementary (New Supplementary Figure 4B).

Referee #2 (Comments on Novelty/Model System for Author):

The manuscript used appropriate methodology and there are no issues ethics or model organisms

Referee #2 (Remarks for Author):

This is an interesting manuscript showing that β-glucan administration affects the outcome of LCMV infection in PAX5-/+ mice.

Major comments

1. Did the authors observe any leukemia development in b-glucan injected mice. B-glucan induces inflammation and this could stimuli leukemogenesis.

We thank the reviewer for bringing up this important point but this was not observed in our studies. Indeed, the potential pro-inflammatory effects of β -glucan have been noted and have to be properly evaluated in the context of each disease as was elegantly discussed in Geller and Yan (PMID: 32760409). In our system, treatment with β -glucan alone for example did not induce any changes in cytokine profiles many of which were not detectable at 15 days post-treatment (data not shown, as evaluated using the ProcartaPlex 19-Plex, for details please see materials and methods section). In contrast, at day 15 post- LCMV infection, levels of pro-inflammatory cytokines and chemokines CXCL9, CXCL10, CCL3 and CCL5 (New Figure 6H) were significantly decreased in the β -glucan pre-treated group. We also checked whether β -glucan treatment alone increased the numbers of B cells in the bone marrow (New Figure 6C) and spleen (New Supplementary Figure 4C) and this was not the case. Proliferation of B cells in the spleen as measured by Ki-67 staining was also not different between β -glucan treated and untreated groups (New Supplementary Figure 4D).

2. Administration of adjuvants is known to induce antibody production. B-glucan acts as an adjuvant driving innate immune responses. Did the authors observe any difference in antibody production upon b-glucan administration?

Both reviewer 1 and 2 pointed out the importance of evaluating the effects of β -glucan on antibody production which we now show in New Figure 6B. Specifically, β -glucan significantly increased LCMV-neutralizing antibody production at Day 30 post-infection. All new results are incorporated and discussed in main body of the text marked in red.

3. Did the authors inject b-glucan after initiation of LCMV infection. This could be more relevant for clinical practice.

We agree that this is a very important and relevant experiment. We therefore injected β -glucan seven days post infection (New Figure 7A). As at day 30 post-infection we saw significant differences in terms of viral titres, neutralizing antibody production, decreased PD-L1 expression on monocytes etc. (Figure 6) in the β -glucan pre-treatment experiment, we chose day 30 post infection to evaluate whether β -glucan treatment post infection alleviated the effects of chronic infection. Viral titres were significantly lower in the liver and spleen and there were improvements in T cell immunity (increased numbers of tet-gp33⁺ CD8⁺ T cells) upon β -glucan post-infection treatment. While there was a trend towards decreased LDH levels 15 days post-infection, neutralizing antibody production was not improved (New Figure 7A-E). A side-by-side comparison of β -glucan pre- and post- infection treatment with a lower dose of LCMV docile (10^4 vs. 10^6) revealed that only β -glucan pre-treatment was able to decrease PD-L1 expression on monocytes in the LN and improve granzyme B expression in CD8⁺ T cells upon restimulation with the LCMV-specific gp33 peptide compared to the infected group alone (New Figure 7F-G). However, KLRG1 expression in CD8⁺ T cells in the liver was increased in both pre and post of β -glucan treatment groups but percentage increases in tet⁺ CD8⁺ T cells were evident only in the former (New Figure 7I). Taken together, while β -glucan administration post infection was able to improve viral titres and improve CTL immunity, the effects were dampened in comparison to β -glucan pre-treatment suggesting an uncoupling from some of the effects on immunosuppressive cells

especially monocytes. All new results are incorporated and discussed in main body of the text in the results and discussion sections marked in red.

4. There is a difference in plasma cells in Pax5^{-/+} mice. This is due to an effect on survival of plasma cells or differentiation of plasmablasts to plasma cells? Is Pax5 expressed in these cells?

PAX5 has been shown to be expressed at relatively low levels in plasma cell subsets compared to other B cells in the bone marrow (PMID: 26187412). We corroborated this finding by extracting data from human PBMCs using The Human Protein Atlas, Monaco dataset and this data is now presented in new Supplementary Figure 3C. We now also present new data showing that the numbers of plasmablasts were no different between Pax5^{+/+} and WT hosts (New Supplementary Figure 3D). As expected, numbers of plasma cells were significantly lower in the BM of Pax5^{-/+} hosts compared to WT (New Supplementary Figure 3D). When we further subdivided the plasma cells based on CD19 expression there were no differences in ratios between CD19⁺CD138⁺B220⁻ (plasma memory) cells and CD19⁻CD138⁺B220⁻ (long-lived plasma memory) cells leading us to conclude that the observed lower levels of plasma cells in Pax5^{-/+} mice resulted from due differences in differentiation of plasmablasts to plasma cells.

Minor comment

1. Figure 1 "of LCMV Docile and (B) survival was monitored" it should be (C)

Thank-you for catching this mistake which has now been corrected.

Referee #3 (Comments on Novelty/Model System for Author):

The model is necessarily a mouse model of PAX5^{+/-} leukemia and although quite preliminary may be extended to childhood BCP-ALL and perhaps augment its prevention

Referee #3 (Remarks for Author):

The article from Lu et al uses a PAX5^{+/-} mouse model to study the responses to certain viral stimuli and the subsequent changes in survival outcome observed after treating animals with a chronic infection. After LCMV infection, these animals do succumb to infection and also reveal a distinct BM environment and cytokine picture.

The authors assess the ability of beta-glucan via an immune training approach to reverse the effects of PAX5^{+/-} animals susceptible to LCMV infection by increasing viral clearance and showing improved T-cell immunity. Interestingly, these animals now do show decreased susceptibility to infection.

The manuscript sits within the Aims and Scope of the journal and is sound and well written.

1. It is unclear from the manuscript exactly what these mice are succumbing to. There is no mention of leukemia in the results? Do the animals have pB-ALL? What

is the phenotype of the B cells and are they clonal? Indeed, why did one animal develop T-ALL? Was that expansion shown to be clonal?

We thank the third reviewer for this comment and agree that the manuscript lacks clarity on the Pax5^{+/+} model. As all our experiments were carried out in the conventional facility (CF) using mice on a C57BL/6J background (backcrossed for at least 10 generations). The previous papers citing leukemia development upon transfer from SPF to CF facility (PMID: 29490943, PMID: 32911536, PMID: 26408659) were carried out in only one specific facility on an unspecified percentage mix of C57BL/6 × CBA background. As the CBA background is more tumor-prone, the lack of B-ALL development can likely be attributed to facility and strain-dependent differences. This is now all explained in the manuscript marked in red. The mice instead succumb to the chronic effects of a persistent infection. This is supported by A) Persistent and significantly higher viral titres in mice that succumbed to chronic infection (Figure 1D), and B) the higher Lactate Dehydrogenase (LDH, which is elevated when tissue damage from for example chronic disease such as an infection occurs) levels in mice that succumbed to the infection (New Figure 1E). Unthymectomized, genetically predisposed mice such as Pax5^{+/+} or TEL-AML1 can be prone to generally clonal T-ALL development when challenged with infectious and mutagenic agents as was previously demonstrated by Dang, J. et al. (PMID: 25855603) and Schindler et al. (PMID: 19570513) respectively, hence likely explaining the one case of T-ALL that developed. This is also now mentioned and incorporated in the manuscript.

2. What is the latency period for PAX5+/- susceptibility to chronic infection as opposed to moving these animals to a conventional facility? Is this period now shorter?

This is another great point which we address with the following statement in the results section of the manuscript where we acknowledge that the mice could have succumbed to infection before any potential B-ALL development:

'In our backcrossed C57BL/6J models, uninfected Pax5^{+/+} mice housed in CF conditions did not develop leukemia within the 16-month observational span. The chronically infected Pax5^{+/+} mice on the other hand succumbed to infection likely preceding any potential leukemia development although none was observed following VSV and acute LCMV infection.'

3. What is the current health monitoring report for this SPF facility - could other viruses already be present?

All experiments were carried out in the S2 conventional facility and mice were bred in the SPF. In the Zentrale Einrichtung für Tierforschung und wissenschaftliche Tierschutzaufgaben (ZETT) where the animals are housed, all breeding rooms are routinely monitored (every three months at least 3 mice are randomly sampled and are consistently negative) for the presence of the following viruses: Mouse hepatitis virus (MHV), Mouse rotavirus (EDIM), Minute virus of mice (MVM), Mouse parvovirus (MPV), Mouse norovirus (MNV), Theiler's murine Enceph. virus (TMEV), Ectromelia virus, LCMV, Mouse adenovirus type 1 (FL) and type 2 (K87), Reovirus type 3,

Pneumonia virus of mice (PVM), Sendai virus using ELISA. For bacteria and fungi, the following are tested for and monitored: Citrobacter rodentium, Clostridium pilliforme, Corynebacterium kutscheri, Mycoplasma pulmonis, Rodentibacter spp., Other Pasteurellaceae, Salmonella spp., β -haem. Streptococci (not group D), Streptococcus pneumoniae, Helicobacter spp., Streptobacillus moniliformis, Staphylococcus aureus, Proteus spp., Klebsiella spp., Pseudomonas aeruginosa, Bordetella spp., Dermatophytes using culture, ELISA and PCR. Parasites tested for include Ectoparasites and Endoparasites-Helminths.

4. Given that the effect of VSV was not enough to compromise these animals, why didn't the authors use an acute strain of LCMV as a direct comparison? Is this response somehow peculiar to VSV?

This is a very important point and we reasoned that an acute infection would be cleared before inducing any long-term effects and would not be enough to perturb the slight immunodeficiency of the Pax5^{-/+} hosts. We explain our rationale in the manuscript as well as show new data. Specifically, we now show that the same dose of the acute LCMV-Armstrong strain (10⁶ PFU) was cleared by both Pax5^{-/+} and WT hosts by day 15 post-infection (New Supplementary Figure 1B) and that survival was not impacted by the acute infection (New Supplementary Figure 1C).

Minor

5. Was the induced cytokine pattern the same for all animals that did or did not succumb to chronic infection? Regardless of B or T cell ALL?

Lactate Dehydrogenase (LDH) was significantly elevated in mice that succumbed to chronic infection (New Figure 1E) but there were no consistently significant differences in cytokine profiles in mice that succumbed to chronic infection.

6. What was the effect on LSK cell numbers after injecting beta-glucan alone into PAX5^{+/-} animals?

There was also a significant expansion of LSK in Pax5^{-/+} hosts and the data is shown in New Figure 5A.

24th Jan 2025

Dear Prof. Pandya,

Thank you for the submission of your revised manuscript to EMBO Molecular Medicine. I am pleased to inform you that we will be able to accept your manuscript pending the following final amendments:

1) Source data: During our standard source data analysis we note that values in several source data files within a figure but also between different figures are duplicated (see attached excel files). We would like to clarify these issues before we proceed with publication of your manuscript. We kindly invite you to check attached source data excel files with identified duplicated values that are color labeled (you can ignore the color scheme) and clarify the cause of these duplications.

2) Figures: Main figures should be uploaded as individual high-resolution files, with their legends at the end of the manuscript file. Please upload all supplementary figures as EV Figures 1-5 also as individual high-resolution files and update their callouts in the main manuscript text. EV figure legends should be placed after the main figure legends in the main manuscript file. Please check "Author Guidelines" for more information:

<https://www.embopress.org/page/journal/17574684/authorguide#figureformat>

<https://www.embopress.org/page/journal/17574684/authorguide#expandedview>

3) In the main manuscript file, please do the following:

- Please address all comments suggested by our data editors listed below:

o Figure legends:

1. Please note that the exact p values are not provided in the legends of figures 1C-E; 2A-E; 3A-D; 4B-D; 5A, 6A-H; 7C, D, F, G, H, I; supplementary figure(s) 2.

2. Please note that in supplementary figures 1A-C; there is a mismatch between the annotated p values in the figure legend and the annotated p values in the figure file that should be corrected.

3. Please indicate what * represents; if this represents p value(s), please indicate the statistical test used and where appropriate, specify the exact p value in the legend(s) of supplementary figure(s) 5A, B".

4. Please note that information related to n is missing in the legends of supplementary figure(s) 2, 5A.

5. Although 'n' is provided, please describe the nature of entity for 'n' in the legends of figures 4B-E.

6. Please note that the error bars are not defined in the legends of supplementary figures 5A-C.

- Add up to 5 keywords.

- Remove "data not shown" (p.7).

- Add "Disclosure Statement & Competing Interests". We updated our journal's competing interests policy in January 2022 and request authors to consider both actual and perceived competing interests. Please review the policy

<https://www.embopress.org/competing-interests> and update your competing interests if necessary.

- Author contributions: Please remove it from the manuscript and specify author contributions in our submission system. CRediT has replaced the traditional author contributions section because it offers a systematic machine-readable author contributions format that allows for more effective research assessment. You are encouraged to use the free text boxes beneath each contributing author's name to add specific details on the author's contribution. More information is available in our guide to authors:

<https://www.embopress.org/page/journal/17574684/authorguide#authorshipguidelines>

- Rename "Materials and Methods" to "Methods".

- In Methods, provide the antibody dilutions that were used for each antibody.

- In Methods, add statistical paragraph that should reflect all information that you have filled in the Authors Checklist, especially regarding randomization, blinding, replication.

- Indicate in legends exact n and exact p values, not a range, along with the statistical test used. To keep the figures "clear" some authors found providing an Appendix table Sx with all exact p-values preferable. You are welcome to do this if you want to.

- Please include structured Methods section that includes a Reagents and Tools Table (should be uploaded as a separate file) followed by a Methods and Protocols section. More information on how to adhere to this format as well as downloadable templates (.docx) for the Reagents and Tools Table can be found in our author guidelines:

<https://www.embopress.org/page/journal/17574684/authorguide#structuredmethods>

An example of a paper with Structured Methods can be found here:

<https://www.embopress.org/doi/full/10.1038/s44320-024-00037-6#sec-4>

- Correct the reference citation in the text and reference list. In the text, a reference should be cited by author and year of publication. Include a space between a word and the opening parenthesis of the reference that follows. In the reference list, citations should be listed in alphabetical order. Where there are more than 10 authors on a paper, 10 will be listed, followed by "et al.". Please check "Author Guidelines" for more information.

<https://www.embopress.org/page/journal/17574684/authorguide#referencesformat>

4) Funding: Rename "Funding" to "Acknowledgements" and make sure that information about all sources of funding are complete in both our submission system and in the manuscript. Currently, Katharina-Hardt-Stiftung, the Christiane und Claudia Hempel Stiftung, and Löwenstern e.V. (FKZ: 3618S32275 and 3618S32274) are missing in our submission system.

5) The Paper Explained: Please provide "The Paper Explained" and add it to the main manuscript text. Please check "Author Guidelines" for more information. <https://www.embopress.org/page/journal/17574684/authorguide#researcharticleguide>

6) Synopsis:

- Synopsis text: Please merge together the blurb and bullet points and upload it as single .doc file.
- Synopsis image: Please format the image to 550 px-wide x (300 - 600)-px high and upload it as a high-resolution JPEG file.
- Please check your synopsis text and image before submission with your revised manuscript. Please be aware that in the proof stage minor corrections only are allowed (e.g., typos).

7) As part of the EMBO Publications transparent editorial process initiative (see our Editorial at <http://embomolmed.embopress.org/content/2/9/329>), EMBO Molecular Medicine will publish online a Review Process File (RPF) to accompany accepted manuscripts. This file will be published in conjunction with your paper and will include the anonymous referee reports, your point-by-point response and all pertinent correspondence relating to the manuscript. Let us know whether you agree with the publication of the RPF and as here, if you want to remove or not any figures from it prior to publication. Please note that the Authors checklist will be published at the end of the RPF.

8) Please provide a point-by-point letter INCLUDING my comments as well as the reviewer's reports and your detailed responses (as Word file).

I look forward to reading a new revised version of your manuscript as soon as possible.

Yours sincerely,

Zeljko Durdevic

*** Instructions to submit your revised manuscript ***

- 1) a .docx formatted version of the manuscript text (including Figure legends and tables)
- 2) Separate figure files*
- 3) supplemental information as Expanded View and/or Appendix. Please carefully check the authors guidelines for formatting Expanded view and Appendix figures and tables at <https://www.embopress.org/page/journal/17574684/authorguide#expandedview>
- 4) a letter INCLUDING the reviewer's reports and your detailed responses to their comments (as Word file).
- 5) The paper explained: EMBO Molecular Medicine articles are accompanied by a summary of the articles to emphasize the major findings in the paper and their medical implications for the non-specialist reader. Please provide a draft summary of your article highlighting
 - the medical issue you are addressing,
 - the results obtained and
 - their clinical impact.This may be edited to ensure that readers understand the significance and context of the research.

Please refer to any of our published articles for an example.

6) Author contributions: the contribution of every author must be detailed in a separate section.

7) EMBO Molecular Medicine now requires a complete author checklist (<https://www.embopress.org/page/journal/17574684/authorguide>) to be submitted with all revised manuscripts. Please use the checklist as guideline for the sort of information we need WITHIN the manuscript. The checklist should only be filled with page numbers where the information can be found. This is particularly important for animal reporting, antibody dilutions (missing) and exact values and n that should be indicated instead of a range.

8) Every published paper now includes a 'Synopsis' to further enhance discoverability. Synopses are displayed on the journal webpage and are freely accessible to all readers. They include a short stand first (maximum of 300 characters, including space) as well as 2-5 one sentence bullet points that summarise the paper. Please write the bullet points to summarise the key NEW findings. They should be designed to be complementary to the abstract - i.e. not repeat the same text. We encourage inclusion of key acronyms and quantitative information (maximum of 30 words / bullet point). Please use the passive voice. Please attach these in a separate file or send them by email, we will incorporate them accordingly.

You are also welcome to suggest a striking image or visual abstract to illustrate your article. If you do please provide a jpeg file 550 px-wide x 300-600px high.

9) A Conflict of Interest statement should be provided in the main text

10) Please note that we now mandate that all corresponding authors list an ORCID digital identifier. This takes <90 seconds to complete. We encourage all authors to supply an ORCID identifier, which will be linked to their name for unambiguous name identification.

Currently, our records indicate that the ORCID for your account is 0000-0003-4794-0035.

Please click the link below to modify this ORCID:
Link Not Available

11) Include a Reagents and Tools Table as part of the Methods section, which can be downloaded from our author guidelines (<https://www.embopress.org/page/journal/17574684/authorguide#structuredmethods>)

Photos 400-800 DPI

*Additional important information regarding figures and illustrations can be found at <https://bit.ly/EMBOPressFigurePreparationGuideline>. See also figure legend preparation guidelines: <https://www.embopress.org/page/journal/17574684/authorguide#figureformat>

***** Reviewer's comments *****

Referee #1 (Remarks for Author):

The authors have made substantial efforts to address the concerns raised in the previous review. These concerns have been now satisfactorily addressed with additional new data presented in this revised manuscript.

Referee #2 (Comments on Novelty/Model System for Author):

The model is adequate. The authors also tested the effect of b-glucan after infection.

Referee #2 (Remarks for Author):

The authors adequately responded to all comments and clarified the effect of trained innate immunity in the induction of adaptive immunity function.

Referee #3 (Comments on Novelty/Model System for Author):

The model is necessarily a mouse model of PAX5^{+/}- but this preliminary work may help the future design of effective prophylactic strategies against chronic viral infections in susceptible hosts.

Referee #3 (Remarks for Author):

The authors have made substantial improvements to their data and the manuscript overall.

The authors addressed the remaining editorial issues.

20th Feb 2025

Dear Prof. Pandyra,

We are pleased to inform you that your manuscript is accepted for publication and is now being sent to our publisher to be included in the next available issue of EMBO Molecular Medicine.

Zeljko Durdevic
Senior Editor
EMBO Molecular Medicine
